# Role of Endothelial Regeneration and Overloading of Enterocytes with Lipids in Capturing of Lipoproteins by Basement Membrane of Rat Aortic Endothelium

**DOI:** 10.3390/biomedicines10112858

**Published:** 2022-11-08

**Authors:** Irina S. Sesorova, Vitaly V. Sesorov, Pavel B. Soloviev, Konstantin Yu. Lakunin, Ivan D. Dimov, Alexander A. Mironov

**Affiliations:** 1Department of Anatomy, Ivanovo State Medical Academy, 153012 Ivanovo, Russia; 2Department of Pathological Anatomy, Ivanovo State Medical Academy, 153012 Ivanovo, Russia; 3Central District Hospital of Noginsk, 142400 Noginsk, Russia; 4Department of Anatomy, Saint Petersburg State Pediatric Medical University, 194100 Saint Petersburg, Russia; 5Italian Foundation for Cancer Research Institute of Molecular Oncology, Via Adamello 16, 20139 Milan, Italy

**Keywords:** endothelial cell, atherosclerosis, regeneration, LDL, enterocytes, lipid overloading, basement membrane

## Abstract

Atherosclerosis is a complex non-monogenic disease related to endothelial damage in elastic-type arteries and incorrect feeding. Here, using cryodamage of endothelial cells (ECs) of rat abdominal aorta, we examined the role of the EC basement membrane (BM) for re-endothelization endothelial regeneration and its ability to capture low density lipoproteins (LDLs). Regeneration of endothelium induced thickening of the ECBM. Secretion of the BM components occurred in the G2-phase. Multiple regenerations, as well as arterial hypertension and aging, also led to the thickening of the BM. Under these conditions, the speed of re-endothelialization increased. The thick BM captured more LDLs. LDLs formed after overloading of rats with lipids acquired higher affinity to the BM, presumably due to the prolonged transport of chylomicrons through neuraminidase-positive endo-lysosomes. These data provide new molecular and cellular mechanisms of atherogenesis.

## 1. Introduction

A significant increase in complications of atherosclerosis in Westerners (after increase in consumption of fat and meat, at least until statins appeared), as well as recent data on the high therapeutic effect of statins, prove the role of cholesterol in the development of atherosclerosis. Hypercholesteremia is dangerous due to the accumulation of lipids in subendothelial space, because it increases the mechanical influence of moving blood over the endothelium [1,2,3]. On the other hand, a rather high incidence of atherosclerosis in vegans (although to a lesser extent than in meat eaters) suggests that atherogenesis is not limited to increased cholesterol intake alone. Moreover, many people consume eggs and meat in high concentrations, and they do not develop atherosclerosis [4,5].

The role of small, dense electro-negative modified low density lipoproteins (LDLs) for the development of atherosclerosis was established by Avogaro et al. [6] and Orekhov et al. [7]. Oxidized cholesterol causes atherosclerosis faster [4]. Of interest, oxidized LDL is not detected in the blood (reviewed by [8]). However, enlarged chylomicrons formed in the lipid overloaded enterocytes circulate in the blood longer and can undergo oxidation [4,5]. In contrast, Tertov et al. [9] concluded that the oxidation of LDLs in blood and food is not important for atherogenesis. The de-sialylation of ApoB48 could be the main atherogenic factor [7,9,10,11,12]. The trans-sialidase activity was found in human blood plasma [11,12,13,14]. However, the sialidase in the blood that was supposed to cause LDL desialylation was not cloned. Desialylated LDLs, but not native LDLs, promoted the accumulation of lipids in aortic intima cell culture [10]. Anti-LDL antibodies bind preferably to desialylated LDLs [15].

The mechanical influence on the endothelium of the elastic-type arteries potentiates the development of atherosclerosis. This is evidenced by the fact that atherosclerotic plaques develop in areas of turbulent blood flow and do not develop in the pulmonary artery. In addition, the development of atherosclerosis is potentiated by hypertension (when the single load on the endothelium increases) and aging (when the number of such single loads increases). The “Response-to-Injury Hypothesis” proposed by Ross [16,17] suggests that atherosclerosis is the result of repeated damage of the endothelium, accompanied by platelet adhesion, their activation on the exposed subendothelial surface, and the migration of macrophages into the intima. This hypothesis postulated that the initiating event in the atherogenic process was some form of overt injury to the intimal endothelial lining, induced by toxic substances or altered hemodynamic forces. Focal endothelial desquamation induces platelet adhesion and the localized release of platelet-derived growth factors, inducing the proliferation and phenotypic modulation of medial smooth muscle cells, thus, generating a fibromuscular plaque [5,18]. In the human aorta fixed with perfusion immediately after death due to mechanical injury, we observed endothelial cells (ECs) with cilia and multinuclear ECs and small areas of endothelial detachment in the areas with turbulent blood flow [19]. However, the specific mechanisms behind this hypothesis were not clear. 

The role of ECs in this process was examined using the endothelial regeneration assay [20,21,22,23,24,25,26,27,28,29,30,31,32,33,34,35,36,37,38,39,40]. Various methods of de-endothelization were used to model the process of endothelial regeneration. These influences include the drying of ECs [41]; using catheters to destroy the endothelium; injecting hypotonic solutions or air into the insulating area of blood vessels; local freezing of blood vessel walls; various chemical [4,5] and mechanical compression of the vessel [42]. The process of regeneration of the aortic endothelium after mechanical damage is described in detail by Schwartz’s and Reidy’s groups [20,21,22,23,24,25,26,27,28,29,30], and the description of similar data after cryoinjury was performed by Malczak and Buck [43] and us [31,32,33,34,35,36,37,38,39,40] more than 35 years ago. However, these older works were not quoted in recent papers studying the regeneration of arterial ECs [42,44].

After cryoinjury, ECs started to detach from their basement membrane (BM) within 5 min after freezing and the subsequent restoration of blood flow, inducing the attachment of platelets and leukocytes to the BM of ECs. After 8–12 h, signs of endotheliocyte migration were detected in the marginal area of the intact endothelial layer. Initially, migrating ECs formed the active leading lamella, which cleaved platelets from the de-endothelialized BM. ECs acquired a streamlined elongated spindle-like shape. Between 13 and 24 h, ECs at the de-endothelialization zone began to synthesize DNA and undergo division. The first 4–7 columns of resident endothelial cells, on both sides of the injury, synchronously entered the cell cycle, propagated, and significantly contributed to the regeneration of the denuded area. By the end of the second day after the injury, a noticeable hyperplastic zone was formed. The mitotic activity of ECs became highest on the third day. Additionally, it was established that ECs adjacent to denuding wounds of rabbit common carotid arteries, both in vivo and in vitro, did not enter the S phase of the cell cycle until between 20 and 28 h after wounding [22]. Altered hemodynamics, especially the decrease in the speed of the intraluminal blood flow, affects endothelial repair [45,46].

Nowadays, interest in the study of regeneration has been revived. McDonald et al. [42] wrote that they discovered that blood flow had no effect on re-endothelialization as both the upstream and downstream fronts of endothelial cells advanced at similar rates; however, this finding was contrasted in 1978 with a balloon-induced EC injury [20] and in 1985 [31] after a cryoinjury of ECs. Itoh et al. [47] stated that they discovered that the recovery of the EC-denuded middle cerebral artery after photochemical injury required a combination of EC elongation, migration, and proliferation from both sides of the injury zone, which was already previously acknowledged in the literature [20,21,22,23,24,25,26,27,28,29,30,31,32,33,34,35,36,37,38,39,40].

Mounting evidence indicates that circulating STEM cells do not directly contribute to endothelial regeneration by forming part of the regenerating endothelium. When resident ECs were damaged via irradiation, bone marrow-derived cells were recruited and incorporated into the injured vasculature, but these cells failed to differentiate into ECs [44]. Endothelial regeneration requires activation of stress response genes [42].

Molecular mechanisms involved in endothelial regeneration are detailed in [42,44]. A table of proteins involved in the restoration of EC integrity is presented by Evans et al. [44]. The regeneration of ECs occurs according to the 2D mesenchymal-type of cell migration; the sub-endothelium becomes thicker. The collective migration of ECs is termed as ‘contact-stimulation of migration’ [48]. When the EC proliferation was suppressed with irradiation, compensating spreading of adjoining endotheliocytes was observed [34]. Repeated lesions also produce an increase in the degree of endothelial heterogeneity and appearance of the multinuclear EC clusters [33]. After multiple freezing of aorta, regeneration of ECs was faster than after mechanical damage of aortic intima because mechanical damage impaired the EC basement membrane (BM) [21,29,33,49]. Importantly, there is no smooth muscle cells (SMCs) in intima because SMCs in media were also killed by freezing [31,35]. No regenerative response was seen in the medial layer, consisting only of elastic and collagen fibers, and its thickness was reduced during the regeneration [2]. Only after complete re-reendothelialization of a large area affected by cryodamage did myo-intimal thickening occur due to SMCs migrating from the edge of the damaged area below the endothelial monolayer from the uninjured edges, and its thickness of SMC layers increased with time [40].

In the current article, we tried to establish a link between endothelial damage, enterocyte overload with lipids, and the development of atherosclerosis. We demonstrated that when enterocytes are overloaded with lipids, glycosylation is disrupted, which can cause the appearance of autoantibodies. This allowed us to put forward a hypothesis that at least partially explained the mechanism of the formation of the modified LDLs.

## 2. Material and Methods

Rabbit polyclonal antibody against CENPF (Anti-CENPF antibody [ab5]) was obtained from Abcam (Cambridge, UK, catalog № ab84697). Polyclonal Anti-PCNA antibody were from Sigma-Aldrich (Milan, Italy, catalog № AV03018-100UG). Anti-cyclin A polyclonal antibody was from Thermo Fisher Scientific (Newark, DE, USA; catalog № PA5-16516). The Invitrogen NEU1 Polyclonal Antibody was from Thermo Fisher Scientific (Newark, DE; USA; catalog № PAS-42552). Invitrogen rabbit polyclonal Antibody against Lysosomal-associated membrane protein 2 (LAMP2) was from Thermo Fisher Scientific (Newark, DE, USA; catalog № PA1-655). Lectin Limax flavus (LFA) conjugated with biotin was from MY BioSource (catalog № MBS656449; in Italy: ITALIA Gentaur SRL, Bergamo). Streptavidin Gold Conjugates (20 nm, 10 OD and 10 nm, 10 OD) were from Abcam (Cambridge, UK, catalog № ab270029 and ab270041, respectively). Rabbit polyclonal antibody against NEU1 was from Thermo Fisher Scientific (Newark, DE, USA, catalog № PA5-42552). KODAK Autoradiography Emulsion, Type NTB, for Micro-autoradiography was from Agar Scientific (in ITALY: Assing S.p.A. [catalog № AGP9284]). Ilford L4 emulsion was from Ilford Ltd. (Ilford, Essex, England). Ilford Nuclear Emulsion Type L4 was from LABORIMPEX (Brussel, Belgium, catalog № AGP9282). For the staining of endothelial monolayers obtained from the rat aorta we used the H&E Staining Kit (Hematoxylin and Eosin) (from Abcam Cambridge, UK, catalog № ab245880) or Hematoxylin and Eosin Stain Kit (from Vector laboratories; H-3502). Polyvinylpyrrolidone (average mol. wt. 10,000) was from Sigma-Aldrich (Milan, Italy, catalog № PVP10). Gum arabic was from Sigma-Aldrich (Milan, Italy, catalog № G9752). Thiocarbohydrazide was from Sigma-Aldrich (Milan, Italy, catalog № 223220). Ketamine HCl (200 mg/mL) was from NexGen (catalog [SKU] № NC-0256). BioGlue^®^ Surgical Adhesive was from CryoLife, Inc. (Kennesaw, GA 30144, USA [BioGlue^®^]. GoldEnhance^TM^-EM was from Nanoprobes, Inc. (Yaphank, NY 11980-9710, USA, catalog № 2113).

Pro-Celloidine for microscopy (4–8% solution of Nitrocellulose in ethanol/diethyl ether) was from Sigma-Aldrich (Milan, Italy, catalog № 81680). The 3 mm diameter copper meshed grids with a holder (Special pattern with handles 100 mesh coordinate Copper) were from Labtech Serving scientists (Heathfield, East Sussex, TN21 8DB, USA, catalog № [KU]: 07D00937). Veco Square Mesh Handle 150 mesh Copper Grids were from Fisher Scientific (Part of Thermo Fisher Scientific; Newark, DE, USA, catalog № 50-289-1). ^3^H-thymidine (specific activity 6.7 FLCi/mmole) was from the New England Nuclear Corporation (Boston, MA, USA) and from Izotop (RF). Samples were taken from 106 rats, namely, 25 normal male WKY rats subjected to standard cryoinjury, 55 male WKY and SHR rats subjected to other more complicated experiments, and 26 normal male WKY rats subjected to correlative light electron microscopy and immuno-electron microscopy labeling. Methodology and the details of all ethics rules are described in [50,51,52]. Briefly, all the experimental animal procedures were approved by the Committees of the Ivanovo State Medical Academy. WKY and SHR rats were obtained from the Moscow Cardiological Center (they took them from Taconic Farms (Germantown, NY, USA) and were maintained either on Purina rodent chow (№ 5001 ICN Pharmaceuticals, Inc., Cleveland, OH, USA) or using the manually prepared food corresponding to the standards. All procedures were in accordance with the EU directive 2010/63/EU.

The animal facilities of the St. Petersburg Pediatric University and Ivanovo State Medical Academy housed animals in plastic sawdust-covered cages on a 12 h/dark/light cycle keeping them under standard conditions of room temperature and fed standard rat pelleted food and water ad libitum. In all experiments, rats were matched for age (6 months), sex (males). Rats were anesthetized with a combination of Zoletil (the active substances, zolazepam hydrochloride and thiamine hydrochloride in equal proportions) and 2% Rometar (the active ingredient was xylazine hydrochloride) in the ratio of 3:1, in a dose of 0.1 mL per 100 g of body weight [50,53].

Animals were removed from the experiment before the end of anesthesia after opening the chest by the intracranial administration of a saturated solution of potassium chloride at a dose of 1–2 mM/kg. Trained persons sacrificed the rats. Death was confirmed when cessation of heartbeat and respiration was observed, as well as the absence of reflexes, in agreement with international standards (https://www.lal.org.uk (accesed on 1 March 2022)). While the animals were under ether anesthesia, jejune tissue was removed, processed, embedded, sectioned, and stained. All experimental animal procedures were approved by the Committees of the Ivanovo State Medical Academy and St. Petersburg State Pediatric University. The procedures for animal use were conducted in accordance with the ethical and legal standards of the Russian Federation mentioned in the Order № 755 of the Ministry of Health of the USSR, 12 August 1977, “On measures to further improve the organizational forms of work using experimental animals”, a letter from the Ministry of Agriculture dated 5 February 2022 № 13-03-2/358, “On modern alternatives to the use of animals in the educational process”, and the 2010/63/EU legislation on animal protection. The experiments were approved by the decision of the Academic Council of St. Petersburg Pediatric University no. 10 from 23 September 2015 and a decision of the ethic committee of Ivanovo State Medical Academy (№ 1 from 5/XII, 2018) in compliance with the above-mentioned details. All experiments on live animals were carried out in Russia; samples were irreversibly fixed with glutaraldehyde, embedded in Epon or gelatin (with subsequent fixation) in Russia and only then were the plastic samples transported to Italy, where these plastic samples were examined. Rats at the age of 24 months were considered by as old animals.

To create DOC-salt hypertension in six-month-old rats, the left kidney was removed, a 1% solution of table salt was given as a drink and 10 mg of deoxycorticosterone acetate (DOC) was injected subcutaneously 2 times a week in the form of 0.5% lipid solution. As a control, animals with a removed kidney were studied, receiving water and injections of appropriate volumes of lipid without a DOC [54,55]. Uni-nephrectomy was performed in 11-week-old WKY rats, and subcutaneous injections of deoxycorticosterone pivalate (1.5 mg/100 g body weight) were administered twice weekly, starting 1 week after uni-nephrectomy. Drinking water contained 1% saline. Tap water and the sodium-deficient diet were given to one group of uni-nephrectomized animals to evaluate the effect of DOC alone for 4 weeks, and to a group of uni-nephrectomized WKY rats for 11 weeks to lower the BP induced by a DOC treatment of 7 weeks. In control rats, the left kidney was removed although DOC and saline were not used. Systolic blood pressure was measured using tail cuff plethysmography with a photoelectric cell detector in a temperature-controlled room at 27 °C.

Cryoinjury to the abdominal aorta was performed in 25 normal male albino WKY rats weighing between 220 and 250 g. Animals were deeply anesthetized by the intramuscular injection of ketamine (50 mg/kg) or as previously described [50]. When the rat started to sleep, the rat’s abdominal cavity was opened and the peritoneum was carefully removed from the surface of the abdominal aorta. The adventitia became open. A droplet of Indian carbon-containing ink diluted 1 to 1 with a double solution of PBS was placed on the abdominal surface of the aortic wall freed from adventitia, for 2 s. Then, the excess of ink was removed with a filter paper. After removing the excess ink with filter paper, the copper probe cooled in liquid nitrogen to −190 °C was carefully, and without any excessive pressure, applied to the aortic wall for 30 s and caused the entire wall to freeze. After this attachment of the probe, the vessel wall under the probe was covered with frost. However, the blood flow was not stopped. The presence of blood flow was controlled by the outflow of blood from the underlying small branches of the abdominal aorta, which were then closed. In the control rats, in which a copper rod applied to the wall had a temperature equal to 37 °C.

The rod was made of a single piece of copper, composed of two parts, each of which had a cylindrical shape. The length of the large cylinder was 30 mm, and the diameter was 12 mm, the length of the lower cylinder, which was directly applied to the vessel, was 10 mm, and the diameter was 3 mm. The surface area of the cylindrical tip of the thin cylinder was equal to 7 µm^2^. At the very end of the thin cylinder, there was a recess in the metal of the cylindrical shape, the curvature of which corresponded to the cylindrical surface of the aorta. The application of the rod to the vessel wall was almost complete: there were no gaps between the wall and the rod. In the control animals, the same probe was applied to the aortic wall, but its temperature was equal to 37 °C. After removing the probe, the frost disappeared within 10 s. After thawing, a second droplet of the Indian ink was carefully placed on the center of the cryoinjured area, and then a speck of powdered carbon was also placed on the vessel. Additionally, two stitches were made on the muscles at the level of the center of the frozen zone exactly indicating the level of freezing of the aorta. Except when the animal was to be sacrificed immediately, the abdomen was closed with sutures and skin clips. In the control rats, a copper rod applied to the wall had a temperature equal to 37 °C.

Multiple cryoinjury was performed as described [33]. Briefly, four cryoinjuries were made every 14 days. For the first cryoinjury, the copper applicator had a base of 6 mm, the second one had a base of 5 mm, and the third probe had a diameter of 4 mm. The final cryoinjury was made by the previously described standard probe, with a work diameter of 3 mm.

Perfusion fixation was performed exactly as described in [35,56]. Briefly, Evans blue (0.5 to 1.0 mL of 1% aqueous solution) was injected into the tail vein 2 h before sacrifice, which was sometimes used as an additional aid in the identification of the injured area. The animal was anesthetized. The abdominal cavity was opened. The right iliac artery was cannulated. The polyethylene cannula was inserted into the vessel against the blood flow. Initially, vessels were washed with mammalian Ringer solution warmed to 37 °C containing heparin (20 units/mL), 0.2%, glucose, and 6% low molecular weight polyvinylpyrrolidone (or 12% bovine serum albumin) under the pressure of 110 mm Hg for 60 s. At the same time, a vein in the axillary fossa was cut. Then, the vessels were perfused for 5 min with the fixative solution containing 2% paraformaldehyde and 2.5% glutaraldehyde in 0.15 M sodium cacodylate buffer (pH 7.4) containing 2 mM calcium chloride at 37 °C for 5 min [9,16]. When immune electron microscopy was applied, the perfusion fixation of the aorta was performed with a mixture of 4% formaldehyde and 0.05% glutaraldehyde on a 0.1 M HEPES buffer (pH 7.4) under systolic pressure (usually 110 mm Hg) for 5 min. In order to acquire the possibility of using the immune EM pre-embedding method, for perfusion fixation we used a mixture of 4% formaldehyde and 0.05% glutaraldehyde in 0.1 M HEPES.

After 5 min of fixation, the abdominal aorta was separated from the tissues with scissors with thin branches; approximately a segment of a vessel 1 cm long, including a 3 mm-freezing zone in the very center, was cut off. Then, with thin branches of eye scissors, the aorta was cut along the back wall so as not to touch the front wall, where the freezing zone was located. After extraction from the rat body, the samples were fixed in the perfusion solution for 1 h. The posterior wall of the aorta was cut lengthwise, and the sample was attached with cactus needles to a thin parallelepiped cut from the cork bark, and then dehydrated and dried by passing through the critical point. Then, these samples were prepared for scanning and transmission electron microscopy exactly as was described [35,50]. A thin layer of gold was evaporated onto the surface of the endothelium. At the same time, in places where the sample had undercut edges, conductive bridges were formed with the help of silver glue, which allowed electric charges to drain onto the sample holder.

Analysis of the LDL attachment to rat aorta was performed on 24 rats. Briefly, LDLs were isolated after the usual standard feeding in the volume of 20 cubic cm and after the introduction of sunflower oil in the amount of 2 mL mixed with crushed standard feed into the stomach. Native LDLs were isolated from plasma by ultracentrifugation as described [57,58,59]. Briefly, LDL (density 1.019–1.063 g/mL) was isolated from the plasma of healthy donors using sequential buoyant density centrifugation techniques, with the use of potassium bromide for density adjustments. LDLs were dialyzed against 0.15 M NaCl. EDTA (1 mM) and butylated hydroxytoluene (0.1 mM) were present during all isolation and dialysis procedures. Additionally, LDLs (1.05–1.02 g/mL) were prepared by two methods, by the rapid procedure of Sattler et al. [60] or by sequential density gradient ultracentrifugation in two KBr gradients by the modified method of Chung et al. [61,62,63].

We overloaded rats with lipids according to Sabesin and Frase [64] with some modifications. In order to overload enterocytes, a thin polyethylene tube was inserted into the stomach of rats that had fasted for 24 h, and 1.5 mL of corn oil to rats or 1.5 g of standard food to control animal, where the lipid content was equal to 10%, was injected there [4,50]. In order to calculate the volume of lipids in standard food we extracted lipids from food with ethanol and then the ethanol was evaporated. We took the volume of sunflower oil used by Sabesin and Frase [64] as a reference portion. Then, we recalculated using this volume in such a way that in the portion of standard food, the amount of lipid was equal to 1/10 of the overloading amount. We observed the enterocytes 2 h after the introduction of food. Normal LDLs obtained from 3 rats that obtained standard food were mixed into one sample. Similarly, LDLs obtained from 3 overloaded rats were mixed into one sample.

### 2.1. Light and Electron Microscopy

A Zeiss LSM510 laser scanning confocal microscope was used for the examination of samples containing the monolayer of the aortic ECs. Samples were examined by scanning and transmission electron microscopy exactly as was described in [35,38,50]. Briefly, the aorta was cut with a fresh razor blade at the level of the middle of the distance between the center of the frozen zone and its edge. Then, with cactus needles, the vessel was pinned to a thin plate of bark and subjected to drying by passing through the critical point. The posterior wall of the aorta was cut lengthwise, and the sample was attached with cactus needles to a thin parallelepiped cut from the cork bark, and then dehydrated and dried by passing through the critical point. A thin layer of gold was sprayed onto the surface of the endothelium. At the same time, in places where the sample had undercut edges, conductive bridges were formed with the help of silver glue, which allowed electric charges to drain onto the sample holder.

After the fixation of the samples, they were post-fixed. Initially samples were washed with 0.15 M sodium cacodylate buffer followed by incubation in the reduced OsO_4_ for 1 h on ice. After washing in distilled water, the samples were incubated again in 0.3% thiocarbohydrazide for 20 min, washed with distilled water, and finally incubated a third time in 2% OsO_4_ in water for 30 min [4,19,20,21,22]. Circles of aorta taken from the control and experimental animals were trimmed in such a way that endothelium from the frozen wall and control wall where copper had a temperature 37 °C were glued and embedded into Epon.

### 2.2. Correlative Light Electron Microscopy

The method of the correlative light-electron microscopy is demonstrated in the scheme shown in (Figure 1). Briefly, the segment of the aorta with the length of at least 1 cm extracted from the surrounding tissues was cut out of the vessel so that the rounded freezing zone (light brown oval in Figure 1A) was in the very center. Then, with scissors with ultra-thin blades, the vessel was cut along the posterior (looking at the spine) wall of the aorta. Then, with the help of cactus needles, the aorta was spread out on a thin cortical plate. The needle was stuck into the wall and fixed in the cortical plate (Figure 1A: a light brown oval is a freezing zone). Additionally, the ends of the aortic wall were glued in the cortical plate using BioGlue^®^ Surgical Adhesive (From CryoLife; Figure 1B: dark red structures). Then, the cactus needles were removed (Figure 1B). A drop of Pro-Celloidine for microscopy was placed on the slide (Figure 1C: structure colored in magenta). A 3 mm diameter copper meshed grid with a holder was superimposed on the frozen segment of the vessel. Then, the glass with a drop of Pro-Celloidine was turned upside down and superimposed on the vessel, as it is shown in Figure 1D. Further, the glass was pressed against the vessel (Figure 1F). This composed sample was placed in 30% ethanol and a cortical plate with tweezers was glued to it by the vessel wall. Next, it was torn off from Pro-Celloidine. A layer of Pro-Celloidine remained on the glass (the yellow layer in Figure 1H). ECs and a copper grid (blue circles above the slide (green plate in Figure 1H)) were embedded in the layer of pro-Celloidine (prismatic structures of blue color in Figure 1H) and also remained attached to the glass. The sample was stained with hematoxylin/eosin, labeled with antibodies and gold, and then prepared and examined under a light microscope (under a layer of liquid so that the cells did not dry out). A number of images of the cell of interest were taken with gradually decreasing magnification to obtain a map of the EC layer. Next, two droplets of 10% gelatin solution in water with a temperature equal to 37˚C were placed onto the EC layer (Figure 1I: gelatin is orange). A small amount of liquid nitrogen was poured on top of the gelatin for cooling (it was necessary to prevent freezing).

Next, the sample was placed in a cold 2% glutaraldehyde solution for 30 min. The grid holder bent up and then pressed against the gelatin. This was followed by the washing from the retainer. Next, a small piece of photographic film washed in a photographic fixer was taken (do not use nitrocellulose film). The fixer was washed. A drop of 10% gelatin solution at a temperature of 37 °C was again placed on the gelatin fixed and washed from the fixative. A piece of photographic film (light green structure) was placed on this drop and pressed tightly against the sample (Figure 1K). The sample was incubated in a mixture of 100% ethanol and 100% ether. Pro-Celloidine was dissolved (Figure 1L). After the severe washout with the same mixture, the sample was dehydrated and then dried by passing through a critical point for viewing in a scanning electron microscope or prepared for examination in a transmission electron microscope (Figure 1M). In the latter case, the drug was initially viewed under a stereo microscope and the cell of interest was on the drug. Then, using a steel needle, two recesses were made in the preparation so that the line connecting them passed through the center of the cell of interest to us. Around our cell, we made several depressions in the formation outside the future analysis zone so that it was easy to find our cell by the pattern of these depressions after pouring into the Epon. Then, the sample was treated with 1% OsO_4_ in water and dehydrated according to the protocol described earlier. Next, our sample was dehydrated and embedded into Epon using its increasing concentrations in 100% ethanol. When the sample had already reached 100% Epon, the sample was placed vertically at a temperature of 60 °C to drain the excess of Epon and preserve the ability to visualize the luminal surface of the ECs [52,56]. Under these conditions, the surface of ECs was visible along with the indentations we had applied to its surface.

After the sample was removed from SEM, it was examined under a stereo microscope and two indentations were made with a metal needle so that the line connecting them passed through the center of the circle where freezing was performed, which was then prepared for examination in the microscope. Then, the samples were removed from the microscope holder and prepared for TEM. After polymerization of the Epon, the cavities were identified and the pyramid was trimmed so that it passed through the middle of the nucleus of our cell into its surface, and the Golgi zone, which was always in the leading edge of the EC, remained in the sample. Next, we cut and looked at semi-thin slices, when the surface area of the section of the nucleus began to decrease and this area became very small, we began to cut ultra-thin slices 250 nm thick, interspersing them with the 60-nm sections. Sections were collected on the slot grids. There, we easily identified the section where the Golgi apparatus began to be sectioned and then serial sections were examined under the electron microscope. When we found the zone where the centriole was located and checked on serial slices, the phase of the cell cycle corresponded to the structure of centrioles.

### 2.3. Scanning Electron Microscopy Radioautography

In order to find ECs in the S and G2 phases, we used the SEM radio autography. The procedure was described in detail in our work [65]. Briefly, under anesthesia (see above), a single pulse of 50 µCi of 3H-thymidine (New England Nuclear Corporation, Boston, Mass, specific activity 6.7 FLCi/mmole) diluted in 2 mL of saline (0.5 mL per 300 g of the animal mass) was injected into the abdominal cavity of the animal. The perfusion fixation was performed 1 h after injection. After washing the fixator in glucose solution, the sample was immersed into the KODAK Autoradiography Emulsion or Ilford L4 melted at a temperature of 37 °C and diluted 1:3 with distilled water for 60 s. Next, the sample was removed from the emulsion and placed vertically on the filter paper so that the excess emulsion could drain. The sample was placed in a closed chamber away from light for 26 days in a refrigerator at a temperature of 4 °C. All this was carried out under a dark red light. The radioautographs were developed in D19b, a well-standardized procedure yielding thread-like silver grains or in the amidol developer for 3 min. Next, our sample was placed in a photographic fixer for 15 min, washed in water and additionally fixed in 2% glutaraldehyde for 30 min, dehydrated in ethanol solutions of increasing concentrations, transferred to 100% ether, and dried by passing through the critical point in a critical point dryer, covered with a thin layer of gold, prepared for SEM, and examined in SEM. After the study in the SEM, the images were placed on a drop of full resin on the cover glass with the endothelium down. Next, the preparation was glued with silver-containing glue to the holder of a scanning electron microscope and conductive bridges were made on the edges of the vessel using the same glue to remove electrical charges from the surface of the sample. After photographing the samples and finding the labeled EC nuclei, successive images were taken under decreasing magnification to obtain a map of the surface of the de-endothelialization zone.

### 2.4. Cell Cycle Determination

Using radioautography, we labeled cells in the S phase with silver track, but instead of the immediate fixation, we waited 3.5 h (the average time of the G2 phase is 3 h [66,67]. Then, cells were fixed. If in 3 h after the application of 3H-thymidine, the isotope and the silver-positive cell remained alone, we considered this cell as potentially being in the G2 phase. Additionally, for the EC monolayer, we labelled the S phase ECs with anti-PCNA polyclonal antibody and G2 phase cells with antibody against CENP-F and cyclin A, the markers of the G2 phase. Then this cell was examined using serial ultrathin sections. During the G2 phase, the previous daughter centriole acquired appendages, and then in each pair of centrioles there was one old centriole with an appendage and a new one without it. Two fully matured pairs of centrioles were in close association with each other. In the end of the G2 phase, just before the mitosis, two pairs of centrioles began to diverge [68,69]. If in this cell, two pairs of fully matured centrioles were observed as being a distance of 50 nm from each other, this cell was considered as being in the G2 phase.

### 2.5. Preembedding Immuno-Electron Microscopy (Pre-IEM)

Pre-embedding was used on the EC monolayer. Samples obtained after correlative light electron microscopy were additionally checked using immunolabeling. The NPG-Silver enhancement was modified from [70]. Briefly, monolayers of aortic ECs were incubated with the solutions used for pre-embedding. Samples were washed from fixative in buffer, incubated with the blocking buffered solution for 1 h, was rinsed four times for 10 min, and then incubated with primary antibody dissolved in blocking solution for 4 h at room temperature. Next, four blocking buffer rinses over 30 min were performed and the secondary antibody incubation was applied, namely species-specific Fab fragments antibody labeled with 1.4 nm nanogold in blocking solution overnight at room temperature and second fixation where 2% glutaraldehyde in 0.1 M sodium cacodylate buffer (pH 7.4) for 15 min was carried out. Further samples were washed in the HEPES buffer: 50 mM HEPES with 200 mM sucrose, pH 5.8, four times over 30 min. Then, incubation in the complete silver enhancer solution from 3–20 min (shielded from light or under dark red light) and the neutral fixer solution composed of 250 mM sodium thiosulfate and 20 mM HEPES at pH 7.4 were applied. To stop the enhancement three rinses were used over 15 min until Gum Arabic was gone. Next, samples were treated with 0.1% OsO4 for 30 min, subjected to dehydration and embedded into Epon.

Pre-IEM based on gold-enhancement was performed according to He et al. [71] with small modifications. Briefly, after the fixation of cells with glutaraldehyde (see above), samples were washed with the blocking buffered solution (four rinses over 30 min), incubated with primary antibody dissolved in blocking solution for 4 h at room temperature, rinsed with blocking buffer (four times over 30 min), and incubated with the species-specific Fab fragments of secondary antibody labeled with 1.4 nm nanogold in blocking solution overnight at room temperature. Then, cells were additionally fixed with 1.6% glutaraldehyde in 0.1M sodium cacodylate buffer (pH 7.4) for 15 min, rinsed with HEPES buffer (50 mM HEPES with 200 mM sucrose, pH5.8, four times over 30 min), washed 3 × 5 min with PBS including glycine (20 mM sodium phosphate, pH 7.4, 150 mM NaCl, 50 mM glycine) to remove aldehydes, rinsed (3 × 5 min) with PBS–BSA–Tween (PBS containing 1% BSA and 0.05% Tween 20), and washed (3 × 5 min) with Solution E (5 mM sodium phosphate, pH 5.5, 100 mM NaCl) from the gold enhancement kit (GoldEnhance-EM 2113).

Next, samples were placed in a mixture of the manufacturer’s Solutions A and B at a 2:1 ratio (80 µL of A and 40 µL of B for 5 min and 200 µL of Solution E with 20% gum Arabic (Sigma-Aldrich), and then 80 µL of Solution C were added in order to develop gold from 7–15 min. The enhancement was conducted at 4 °C. Further, samples were transferred to the neutral fixer solution composed of 250 mM sodium thiosulfate and 20 mM HEPES at pH 7.4 to stop the enhancement (three rinses over 5 min), washed with buffer E for 3–5 min, incubated in 1% OsO4 in 0.1 M sodium phosphate (pH 6.1) for 60 min, and rinsed with distilled water. Finally, after standard dehydrations, cells were embedded into Epon. Semi-thin sections were cut, achieving an exit to a depth of no more than 5 microns because only at this depth antibody solutions penetrate cells. Immunolabeling was determined at a depth of up to 5 microns from the cut surface. After finding a cell with immunolabeling, serial ultrathin sections were made, and the cell phase was determined based on the number of centrioles.

### 2.6. Sampling

When comparing control and experimental animals (for example normotensive and hypertensive rats; 6-month and 24-month rats, rats with single or multiple re-endothelialization) of the appropriate age, randomly selected pairs of six pairs were kept in a common cage, scored at the same time and samples were carried out simultaneously. Each experiment contained one control and one experimental rat. Six pairs of animals composed of control and experimental rats, namely, WKY and SHR rats at the age, the control rat with one cryo-destruction and the experimental one with four cryo-destructions. The rat pairs were sampled on different days. The thickness of the subendothelial zone was calculated 14 days after the last de-endothelialization. We did not consider the thickness of the BM after re-endothelization in hypertensive and old rats, since the increase in thickness would be insignificant. The BM was already thickened by repeated injuries associated with hypertension and aging. However, the BM increased its thickness before cryoinjury by 12% after the single re-endothelialization (our unpublished results). Two measurements or complete examination of two random serial sections of the same cell were performed. For the SEM analysis, we used standard control groups. Estimations of stereological parameters were performed as it was described in [72,73].

Calculation of the percentage of cases when there were multinucleated cells or leading ECs in the endothelial regeneration zone was carried out on en-face monolayer preparations. In the re-endothelialization zone or on the edge in the leading section, 3 or 4 sections were randomly selected. If there were multinuclear ECs or leading ones, then the sample was considered as positive. If such cells were not observed, the sample was negative. Then, the average percentage of positive samples was calculated for the experimental and control (in which the copper probe had a temperature of 37 °C) animals. These mean values were considered as the statistical units. The numbers of units were equal to 6 in the experimental and control groups (paired sampling). This average percentage was higher in a rat with a regeneration zone. Statistically, there was no need or opportunity to compare the controls and experiments since there was simply no such zone in the control animals (the copper rod had a temperature equal to 37 °C). Therefore, the percentage of such sites was statistically compared after the single cryoinjury, multiple cryoinjuries, and single cryoinjuries in hypertensive and old rats. These data were also checked on samples prepared for examination in the scanning electron microscope, which turned out to be similar (these unpublished data of ours are not given).

The volumetric density of mitochondria in the zone of re-endothelialization after cryopreservation and in the control zone in the control animal (the place to which the copper rod was applied with a temperature of 37 °C) was carried out on random sections perpendicular to the axis of the vessel, using classical stereology [73]. The volumetric density of mitochondria *v*/*v* mx (%) was estimated using a squared stereology test-grid lattice; *v*/*v* mx was equal to the number of test-points over mitochondria (Tmx) divided on the number of test-points over ECs (Tec) and multiplied by 100.

The calculation of the EC volume was carried out as follows: on random images obtained in a scanning electron microscope from a sample completely spread out and located at an angle of 90 degrees to the electron beam, the number of ECs was calculated using the stereology method. The method is illustrated in Figure 1N. First, the number of ECs that were completely included in the frame was counted, then the number of ECS that crossed the upper and right boundaries of the parallelepiped was counted. The smallest sections of the ECs were also included in the calculation. The cells that crossed the left and bottom sides were not taken into account. In Figure 1N, the number of ECs completely caught in the frame is 18, the number of ECs crossing the upper and lower sides of the image is 23.

The pore size in the BM was compared in rats in the re-endothelialization zone after a single injury. The pore size in the BM in hypertensive and old rats after cryopreservation increased slightly, and in many cases, it was not possible to gain statistically significant figures due to heterogeneity (not shown).

### 2.7. Statistical Analysis

Statistics were performed using GraphPad Prism 9.4.0. The number of rats used for each analysis is indicated in the figure legends, selected by a power analysis to detect a 30% change with 10% error and 95% confidence. For the sake of simplicity, Student’s *t*-tests, paired *t*-tests, and non-parametric Mann–Whitney *U* tests were used. In the majority of cases, we used the nonparametric Mann–Whitney *U* test. A difference was considered significant when *p* < 0.05. For comparison of 2 groups of continuous variables with normal distribution and equal variances, 2-tailed unpaired Student’s *t* tests (with additional Welch correction for unequal variances) were performed with a significance threshold of *p* ≤ 0.05. In most cases, data are given as the mean ± standard deviations (SD). Values are mean ± SD of 6 variants (*n* = 6). In the text, the words “differ”, “smaller”, or “higher” indicate that two values are significantly (*p* < 0.05) different [74].

## 3. Results

Most of our works on regeneration of the aortic endothelium after cryoinjury were published before 2000 and are not available to the ordinary readers; therefore, we have described these data very briefly and provided the main images reflecting these issues. For instance, SEM of ECs of rabbit aorta was well-described [75]. We have already described some features of ECs in hypertensive and old rats [76,77]. Previously, we have already noted the thickening of the ECBM in the aorta of old rats [78,79]. Here, we examined specific features of the EC regeneration after multiple cryoinjury in old and hypertensive rats. However, these works are difficult to find now. Here, we added some new features and show some images obtained during preparation of the previous articles.

In control animals (see Methods), we did not find any alteration in the ECs in the zone where the copper probe with the temperature equal to 37 °C was applied (Figure 2A). After elimination of the probe, the frozen vessel wall was warmed by the blood flow, the ECs in which the plasma membrane was perforated with ice crystals were separated from the vessel wall and exfoliated (Figure 2B). Platelets were attached to the exposed BM of the endothelium, which had pores (see below) and was flattened out (Figure 2C,E). The leukocytes were visible among the platelets (Figure 2D). There was no detachment of ECs and attachment of platelets in the zones adjacent to the frozen area (not shown). Then, the ECs located on the edge of the damage began to spread out. Their leading edge cleaved off platelets (Figure 2F). The leading lamellipodium formed by spreading ECs contained actin bundles (Figure 2G). Further, 24 h after the single cryoinjury, the ECs began to replicate DNA and divide. Figure 2I shows the labeled ECs in the S phase. In the control, radioautography labeled rare single ECs or did not label them at all (Figure 2H). ECs acquired a spindle-like (or fusiform) shape. A hyperplastic zone was formed (Figure 2J). On the cross-section, the ECs have a bell-shape and protrude into the lumen of the vessel (Figure 2K).

Three days after the single cryoinjury, the de-endothelialized area remained (Figure 1O). The average length of the de-endothelialized zone and the thickness of the endothelial layer in the re-endothelialization zone was significantly (*p* < 0.05) higher than in the control samples and in the areas proximal and distal to the frozen zone (Figure 1P: there the bar “Single” is significantly (*p* < 0.05) higher than the bar “CTR”). The EC volume in the re-endothelialization zone significantly (*p* < 0.05) decreased from 175 µm^3^ to 70 µm^3^ (Figure 1Q). The volumetric density of mitochondria in ECs significantly (*p* < 0.05) increased 2.5-fold (Figure 1R). The complete re-endothelialization of the cryo-damaged area was observed in fourteen days.

The method of en-face film preparations shows the presence of mitotically dividing and multinuclear ECs in the hyperplastic zone (Figure 3A and Figure 4A). Arrows in Figure 3A show cells in the prophase. Arrows in Figure 4A show mitotic ECs formed after a single cryotherapy. Occasionally, individual ECs broke away from the uniform leading edge and formed leading spindle-shaped cells (Figure 3B). The percentage of areas where attached leucocytes were visible increased significantly (*p* < 0.05; Figure 1S). ECs moving in front of the regenerating endothelium are shown by arrows. Importantly, after multiple cryoinjuries (the “Multiple” bar) and in hypertensive rats (“DOC” bar) the percentage of leading ECs was significantly (*p* < 0.05) (Figure 1T) higher than after the single cryoinjury (see below).

The correlation of the scanning electron microscope (SEM) with radioautography and our method of en-face EC monolayer (Figure 3F,G) allowed us to visualize ECs in the G2 phase (Figure 3F–K) and the surface of mitotic ECs at different stages of mitosis (Figure 4A–G). Blebs and short microvilli were formed on the cell surface of the mitotic Ecs (Figure 4B–H). Figure 3H demonstrates five ECs labeled with silver granules (white dots). White arrows show the mitotic ECs and the red arrow shows the EC in the G2 phase. In Figure 3I, the same area is shown on the en-face monolayer preparation. White arrows show the mitotic ECs and the black arrow shows the EC in the G2 phase. Figure 3J demonstrates the pattern of labeling for the CENPF marker of the G2 phase. Figure 3K shows the mitotic EC labeled with silver granules under SEM.

The contacts between the mitotic EC and the surrounding ECs were wider than between normal ECs. This is clearly visible on the tangential section of mitotic ECs (Figure 4H). In normal aorta, the percentage of ECs in the S phase (silver labeling after the SEM-based radioautography) was equal to almost zero. Three days after a single cryoinjury, it was equal to 18 ± 4% in the re-endothelialization zone. After multiple injuries, it significantly (*p* < 0.05) increased to 37 ± 5% in the re-endothelialized zone formed three days after the injury. Fourteen days after the multiple cryoinjury, the thickness of the subendothelial space could reach 18 µm in the zone where subendothelial leucocytes were observed.

After identification of the G2 phase ECs in the en-face preparations (Figure 5), we performed serial sectioning and checked our conclusion that this cell was in the G2 phase of the cell cycle using the pattern of centrioles. Figure 5A–C demonstrate centrioles in the G1 (Figure 5A,B) and S phases (Figure 5C). Figure 5D shows the presence of two pairs of centrioles at the distance more than 500 nm. This confirms that this EC was in the G2 phase. In the G2 phase ECs, we observed the presence of the endoplasmic reticulum exit sites (Figure 5E,F). Moreover, in such cells, we observed distensions of Golgi cisternae filled with collagen-like fibrillar material (Figure 5G). Post-Golgi carriers filled with the procollagen-like fibrillar material was also observed (Figure 5H). Immuno electron microscopy based on pre-embedding revealed that these post-Golgi carriers were labeled for LAMP2, a marker of late endosomes and lysosomes (Figure 5I,L).

Previously, we demonstrated that after re-endothelialization, the thickness of the ECBM increased, and this could explain that after the second round of re-endothelialization, the speed of the EC regeneration increased (see Introduction). Therefore, we examined the role of such BM for EC regeneration. Initially, we studied the alteration of the EC surface after multiple cryoinjuries, and in hypertensive (Figure 5K) and old rats (Figure 6A–F). After multiple cryoinjuries, the percentage of samples, where the attached leucocytes were observed, increased (Figure 6B) in comparison with the single cryoinjury (Figure 6A; quantified in Figure 1S). Additionally, multiple cryoinjuries induced a higher percentage of cases with aggregates of platelets in the de-endothelialized zone (Figure 7B: quantification was not shown) and the formation of multinuclear ECs (Figure 3C–E and Figure 6C,E; quantified in Figure 5N) in the re-endothelialized zone.

After multiple cryoinjuries, the regeneration of SMCs were observed, but these SMCs moved to the injured zone within the subendothelial zone, but not within media (Figure 5M,O). Figure 5J shows the movement of SMCs from media into intima at the edge of the injured zone. Multinuclear ECs were observed within the regeneration zone after multiple injuries (Figure 3C) and after the regeneration of ECs in hypertensive (Figure 3D and Figure 7C) and old (Figure 3E and Figure 7E) rats. Alteration of the surface of the spindle ECs (Figure 7G) isolated leading ECs (Figure 7F), and the unusual shape of the de-endothelialized zone were observed after multiple cryoinjuries (Figure 7H) and in hypertensive and old rats (not shown).

Further, the percentage of areas with leucocytes attached to the EC surface increased in hypertensive and old rats (Figure 6C,D; quantified in Figure 1S). Multiple cryoinjuries (Figure 6B), hypertension (Figure 6D) and senescence (Figure 6C) led to the increased of the percentage of areas with leucocytes attached to the surface of ECs and aggregates of platelets (quantification is not shown). Hypertension induced well defined alteration of the luminal surface of endothelium (Figure 6E,F). Quantitation of these data are presented in Figure 1O–T. The TEM section of this zone is presented in Figure 7A.

Pores in the ECBM ECs (Figure 8A) became narrower after the single cryoinjury (single round of de-endothelialization–reendothelialization; Figure 8B; quantified in Figure 9J). After multiple cryoinjuries, the thickness of the BM-like layer located under ECs increased to an even greater extent (Figure 8D). Increased thickness of this layer was found in old (Figure 8E) and hypertensive rats (Figure 8C). After repeated cryoinjuries (Figure 8H), as well as in old (Figure 8F) and hypertensive (Figure 8C) rats. In all these cases, also, mononuclear leucocytes were observed below ECs (Figure 8C,G: hypertensive rat; 5F: old rat; 5H: multiple re-endothelialization), although a small BM thickening was observed even after (Figure 8B; compare with control in Figure 8A). Additionally, accumulation of leucocytes was observed below ECs (Figure 8C,F,H,J–L). The speed of endothelial regeneration was higher in rats with hypertension, old rats, and after multiple de-endothelialization (Figure 1T). Accumulation of the BM-like material was accompanied by the significant (*p* < 0.05) decrease in the BM pore diameter (Figure 9A–I; quantified in Figure 7L). The BM was clearly visible in TEM sections and after the dissolution of cell membranes (Figure 9H,I).

Earlier, we showed that in newborn rats, when the GC is overloaded, the chylomicrons were delayed in the post-Golgi compartment positive for LAMP2, a marker of lysosomes and late endosomes [80]. Since we were familiar with the contradiction of desialilyation of modified LDLs [5], we checked whether in adult animals, after the enterocyte overloading with lipids, post-Golgi carriers pass through LAMP2-positive compartments. To test the hypothesis that desialylation of ApoB and superficial lipids of chylomicrons may take place due to alterations of post-Golgi transport, we studied enterocytes of the small intestine taken from rats that were fed with a large amount of lipids into the stomach. After 30 min, samples were taken in accordance with the recommendations by Sabesin and Frase [64].

Under normal conditions, post-Golgi carriers positive for LAPM2 were not found (unpublished results). However, earlier, we found that when the enterocytes of newborn rats were overloaded with lipids, the large chylomicrons were accumulated in the LAMP2-positive post-Golgi compartment [80]. The overloading of enterocytes with lipids led to an increase in the size of chylomicrons (Figure 10D–I quantified in Figure 11I) and their prolonged stay in the post-Golgi compartments, which in 33% of cases were marked for LAMP2 and Neu-1. (Figure 10A,B,D,E). Under these conditions, part of the LAMP2 labeling leaked and turned out to be on the basolateral plasmalemma (Figure 10D). Using correlative light electron microscopy, we observed chylomicrons within the LAMP2-positive compartment in 4 enterocytes form 5 such cells examined. The labeling of enterocytes for sialic acid using gold-conjugated lectin from Limax flavus showed that after lipid overload, the labeling of chylomicrons located between enterocytes was lower than when feeding standard food of the same weight (Figure 10E,G–J).

Next, we tested whether the overloading of enterocytes with lipids would induce higher attachment of LDLs to the BM of ECs. Initially, we compared the ultrastructure of enterocyte chylomicrons after new types of different modes of lipid overloading (see Methods). After feeding with a small amount of lipids, the situation was fully compatible with our description presented in the previous paper in [50] (Figure 10A–C). Small chylomicrons were passed through the Golgi complex being accumulated in cisternal distensions (Figure 10A) and were then transported towards the basolateral PM in post-Golgi carriers (Figure 10A). Then, small chylomicrons were accumulated between enterocytes and then appeared in the interstitial space (Figure 10B) and inside the lumen of lymphatic capillaries (Figure 10C). After rat overfeeding with lipids, the Golgi complex was overloaded with larger chylomicrons (Figure 10D). Then, large chylomicrons were observed between enterocytes (Figure 10E) and in the interstitial space and inside the lumen of lymphatic capillaries (Figure 10G). In the control rats fed with a small amount of lipids, labelling for LAMP2 was not observed within the post-Golgi carriers (Figure 10F,I). In contrast, after lipid overloading the larger chylomicrons (quantified in Figure 11I), larger carriers labeled for LAMP2 (Figure 10I) and Neu1 (Figure 10J) were observed.

The overloading of enterocytes increased the ability of newly formed LDLs to bind to the EC nude BM. Incubation of the nude BM visible after a single cryoinjury with LDLs obtained from rats fed with a high amount of lipids induced higher labeling of the BM with LDLs (Figure 11B,D) than with LDLs obtained from the control rats fed with a small amount of lipids (Figure 11A,E: quantified in Figure 11K). Additionally, extensive synthesis of the BM by ECs alter the binding of LDLs to the BM. After incubation of the nude aortic endothelial BM visible after the single cryoinjury with the Limax flavus lectin, the level of the lectin attachment was significantly (*p* < 0.05) lower than after incubation of the lectin with the nude BM formed after multiple cryoinjuries (Figure 11E–G: quantified in Figure 11J).

Thus, LDLs isolated from normal rats had higher affinity to the BM formed after multiple re-endothelialization compared with after a single one. The concentration of sialic acids in the thick BM was lower. Furthermore, lipid particles obtained from rats subjected to lipid overloading with lipids had higher affinity to the normal BM.

## 4. Discussion

In our study, we used the cryodamage of ECs of rat abdominal aorta proposed by Malczak and Buck [43] and modified by us [31,32,33,34,35,36,37,38,39,40]. This method allowed to preserve the endothelial BM and obtain a very uniform edge of the regenerating endothelial front. Initially, we reproduced the regeneration of the aortic endothelium after cryopreservation as a control. The sequence and phenomenology of the extraction was the same as what we had published earlier. Additionally, we have illustrated this process with quantitative data. We established that re-endothelialization led to augmentation of the BM thickness and reduction in the BM pores. ECs of the re-endothelialization zone became thicker, whereas the volume of ECs there decreased. In contrast, the volume fraction of mitochondria in these ECs increased.

This thickening suggested that the ECs synthesized the components of the BM. Our analysis of ECs in different phases of the cell cycle revealed that ECs in the G2 phase exhibited evidence that this occurs during this phase. These ECs exhibited the exit sites from the ER. This indicates that ECs had the active intracellular transport. Also, there were distensions of cisternae in the GC filled with fibrillar contents and the post-Golgi carriers with fibrillar components. LAMP2 is detected in these carriers and in late endosomes, which are often labeled with LAMP2, and might contain Neu1 [5].

Next, we checked this using hypertensive and old rats. When a rare event was repeated a greater number than once (aging) or when its frequency increased, as with hypertension, the adhesiveness increased even more. First, we checked whether there were similarities in the phenotypic changes in ECs in repeated cryoinjuries and in hypertension and aging. We found that after repeated re-endothelializations with hypertension and aging, there were more leukocytes per EC, the thickness of the subendothelial layer was also higher, and leukocytes accumulated in the subendothelial layer. The rate of re-endothelialization with repeated injuries in both hypertensive and old rats was higher than in normal rats. In all these cases, platelet aggregates were more often seen. A thick BM and a decrease in the size of the pores in the BM can affect its adhesion to LDLs, which can penetrate through open interendothelial contacts around mitotically dividing ECs that occur in areas of turbulent blood flow. We isolated LDLs and found that they attach better to the BM after a single re-endothelialization. If this is the case, then an even greater thickening of the BM during repeated re-endothelialization should lead to greater adhesion of normal LDLs.

Importantly, in the human aorta, at the top of the flow divider of the ostia of intercostal arteries, stellate ECs with dilated intercellular clefts could often be seen in those zones, which were devoid of any de-endothelization [56]. Signs of endothelial regeneration were found in the aorta in healthy people at the sites of blood flow division at the mouths of intercostal arteries and at the top of the emerging atherosclerotic plaques. Importantly, SMCs from undamaged zones migrate into the subendothelial layer forming spindle-shaped protrusions with their tip directed towards the cryo-damaged zone, which is already covered by regenerated endothelium [5,41].

The multiple re-endothelialization and arterial hypertension and aging led to the more prominent thickening of the BM. Additionally, senescence and hypertension induced hypertrophy and hyperplasia of the arterial ECs [81,82,83]. The intima within flow dividers resembles fibrous cap of the plaque [84]. Moreover, after the replacement of laminar blood flow in abdominal rat aorta with a turbulent one, no atherosclerosis is observed in rats, whereas the intimal thickening is formed [85]. Although the presence of endothelial progenitors is shown [86], participation of these progenitors in re-reendothelialization in the process of regeneration of the aortic endothelium has not been proved [42]. 

In our hands, the thickening of the BM of ECs seemed to be higher than it was presented by Haudenschild et al. [82]. However, we measured the maximal thickness of the BM. In our hands, after re-endothelization, the affinity of the BM to LDLs became higher than in normal rat aorta. The thick endothelial BM captured more LDLs. Normally, LDLs were not found in this region [87]. LDL transport from the circulation to the vessel wall is promoted at sites of disturbed flow because of flow stagnation and the subsequent prolonged contact between blood and vascular endothelial cells [88,89,90,91]. The LDL binding is most extensive in the arterial branch points where the elastin layer is absent, indicating that the absence of an elastin layer contributes to the initial LDL infiltration at these sites and the subsequent development of atherosclerosis [91,92].

### 4.1. Role of the Enterocyte Overloading with Lipids

Previously, we showed that when overloading Caco2 cells that have been differentiated into enterocytes similar to those found in the small intestine, the glycosylation of ApoB is disrupted [93]. In addition, we found that the phenomenon of enterocyte overload occurs in newborns at the first feeding and the enlarged chylomicrons pass through the post-GC compartment, positive for Neu1, and presumably, therefore, for LAMP2. Therefore, we decided to check whether ApoB desialylation occurs when enterocytes are overloaded with lipids. We repeated the experiments using the Sabesin method and the Phrase and found the signs of overload described earlier [50]. If, normally, post-GC carriers are not marked for LAMP2, then when overloaded, enlarged chylomicrons stagnate in compartments positive for LAMP2 and Neu-1. In addition, there is a decrease in the content of sialic acid on the ApoB of large chylomicrons. We isolated LDLs after normal feeding and after lipid overload and found that LDL obtained in the second case had greater adhesion to the BMEC of the aorta than LDL isolated from rats with normal nutrition.

It is known that when enterocytes of the small intestine are overloaded with lipids, chylomicrons become larger and glycosylation errors occur [4,50,92,93]. Due to the overload of the Golgi complex with large chylomicrons, blood group antigens appeared on glycosylated chylomicrons, which normally should not be synthesized [93]. When the rabbits were fed with cholesterol for 30 days once a day or twice a day, the area of the aorta stained with Sudan III was smaller in the second case [93,94]. Additionally, it was proposed that the atherogenicity of LDLs is linked to the ability of its ApoB moiety to interact with arterial wall proteoglycans [5,95,96]. Indeed, initially formed atherosclerotic plaques contained the neutral complex biantennary glycan, which is a ligand for a macrophage asialoglycoprotein receptor that is involved in foam cell formation [97]. The overloading of small intestine enterocytes with lipids induced the formation of chylomicrons with a lower concentration of sialic acid. Chylomicrons move through endosomes positive for neuraminidase and stay there for a longer time. The long stay of chylomicrons in this compartment leads to the cutting off of the remnants of sialic acid and de-sialylation. Chylomicrons are subsequently transformed into LDLs [4,5,98,99].

LDLs formed after the overloading of rats with oil acquired higher affinity to the BM, presumably due to the prolonged transport of chylomicrons through neuraminidase-positive endo-lysosomes and desialylation of ApoB. It has been suggested that lipoproteins, after their formation and ingestion into the blood, undergo desialylation by sialidases that circulate in the blood [11]. The following four types of mammalian sialidases have been described: Neu1, Neu2, Neu3, and Neu4, which are encoded by different genes and are characterized by different subcellular localizations [100]. Additionally, it is possible that the thick BM captured more LDLs because this BM had a lower number of pores.

This accumulation of large chylomicrons in the LAPM2 and Neu1-positive compartments could induce their desialylation. At the same time, we described the structure of ECs during mitosis and found that the interendothelial contacts of mitotic cells with their surrounding neighbors expanded. This phenotype is also found in humans in the areas with turbulent blood flow [56,101]. In the zones with laminar blood flow, the overall basal rate of the EC replication is nearly negligible [102,103]. In our experiments, when re-endothelialization occurred over the thick BM it became faster. In the areas of turbulent flow, re-endothelization was faster [36]. In contrast, when the model of the EC regeneration based on the mechanical damage of the endothelial BM was used, the regenerative response of ECs was significantly impaired in aging mice due to the reduction in Atf3 expression [42]. However, if one took into consideration that in order to migrate, ECs need to secrete BM, everything becomes clear. The reason is simple, namely, ECs in their model, in order to migrate, need to restore their BM; in contrast, in our model, the BM was not damaged. In addition, the speed of re-endothelialization was faster because the thick BM had smaller pores and could better meet the needs of the migrating ECs.

### 4.2. Hypothesis

On the basis of our data, we proposed the following hypothesis of atherogenesis. The overloading of enterocytes with plant-derived oil would induce the delay of the chylomicron transport in their post-Golgi compartments containing enzymes involved in desialylation. Under these conditions, chylomicron became large and for a longer time circulated in the blood, which could induce lipid oxygenation. Although vegetarians eat no cholesterol, their enterocytes use cholesterol synthesized by the liver and delivered by blood. The endothelium of elastic arteries, located in areas of turbulent blood flow, is damaged mechanically. In the areas of turbulent blood flow in arteries of elastic-types, endothelial cells are subjected to damage and their barrier function decreases. Desialylated LDLs penetrate the endothelium in these arteries and are captured by the basement membrane of endothelial cells. This activates ECs and induces the penetration of monocytes into the intima and the consumption of LDLs and their oxygenation in their lysosomes, resulting in the formation of foam cells. Collectively, our data link together these hypotheses of atherogenesis based on the diet-influenced alteration of lipid metabolism, the EC damage, and the LDL desialylation, and explain why atherosclerosis is observed in vegetarians not eating cholesterol. The scheme describing EC regeneration after cryoinjury is presented in Figure 12A. The scheme of the SMC regeneration after cryoinjury is shown in Figure 12B. The scheme demonstrating the augmentation of the thickness of the subendothelial layer is shown in Figure 12C–E).

### 4.3. Future Perspectives

This article did not solve all problems related to the pathogenesis of atherosclerosis. It represents only the first step in a long road toward the goal. It could be just the beginning of a new, large cycle of research on the mechanisms of atherogenesis. It became clear that desialylation of chylomicrons, and then the LDLs formed from them, leads slowly but surely to atherogenesis and it does not matter whether a person consumes cholesterol or not. If there is no cholesterol in the food, then enterocytes receive cholesterol from the blood, since the liver synthesizes it in large quantities. It is necessary to check in vitro where post-Golgi carriers are delivered to the center or to the periphery. Although there is no pulsating blood flow in the test tube, it may be important to repeat in vitro experiments with the EC freezing and multiple cryoinjuries and check whether they will regenerate faster. Additionally, it is plausible to study the role of limiting the consumption of vegetable lipids at a time. Studies should be conducted in vegans (also, they should eat only fresh food). It could be useful to examine the composition of BM on ECs with cell culture or in a three-dimensional gel, which is formed normally, during repeated regenerations and during prolonged cultivation, simulating aging. To confirm our data, that the synthesis of BM is mainly in the G2 phase, we used cell culture and performed the search of molecular mechanisms. It could be interesting to check whether large chylomicrons could enter the bloodstream or if it is filtered in the regional lymph node by lymph node macrophages. It is necessary to check in an experiment, whether fractional feeding of rabbits with cholesterol leads to a decrease in the severity of atherosclerosis. It is necessary to study in more detail the composition of LDL after overfeeding and after regular feeding. It would be interesting to clarify whether large chylomicrons get into the lymph and the lymph vessels and take the lymph from there.

On the other hand, human intima contains cells such as SMCs, pericytes, and so-called stellate cells, which differs from that of small laboratory animals in which the intima contains only the basement membrane. In large animals, atherosclerotic lesions resemble those in humans. Small animals, namely, mice, tree shrews, and rats, are less sensitive to atherosclerosis than men and monkeys [16,17]. Giraffes with a well-developed aortic intima and high pressure, indicating a strong mechanical influence on ECs, do not have atherosclerosis (only arteriosclerosis was observed, reviewed by [4]). Feeding of large animals with cholesterol for a long time could clarify many issues. One of possibly interesting objectives for future analysis could be hyenas, which eat a lot of oxidized cholesterol. Maybe they have special adaptations against the accumulation of large chylomicrons loaded with oxidized cholesterol. For instance, there could be a special structure of the small intestine or the influence of high level of androgens [104,105].

## Figures and Tables

**Figure 1 biomedicines-10-02858-f001:**
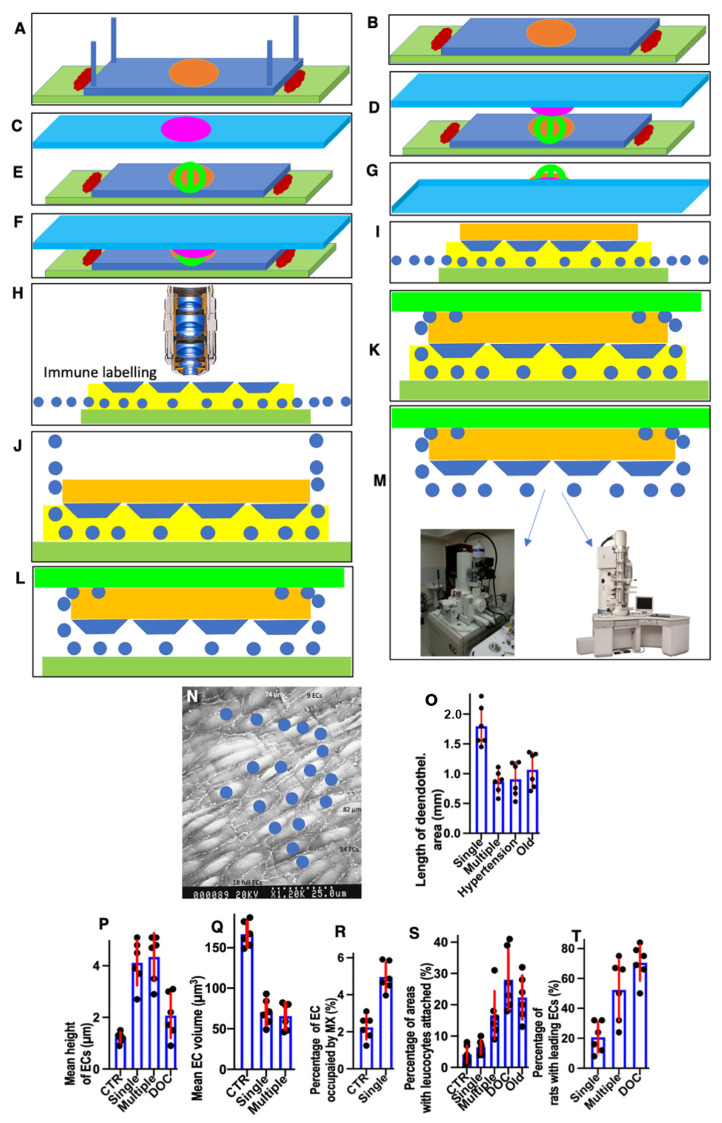
Scheme of the correlative light-electron microscopy of ECs on the basis of the en-face monolayer. (**A**–**M**) The steps of a sample preparation are described in “Material and methods” Section 2.2. (**N**) Scheme demonstrating the method of the EC counting under SEM. Example of our sampling illustrates the method of the EC volume estimation. The ECs, the border of which were not crossed by the sides of the image, are indicated with blue rings. Numbers of ECs that are crossed by the right and the upper side are indicated on the figure. The total number of ECs taken into counting is equal to 43. The surface area of the image is equal to 74 µm × 82 µm = 6068 µm^2^. The surface area of the projection of one EC is equal to 6068/43 = 141 µm^2^. Taking into consideration that the mean thickness (height) of EC is equal to 1.13 µm, the mean volume of ECs becomes equal to 160 µm^3^. (**O**) Length of de-endothelialized zone observed 3 days after the single cryoinjuries (Single) is significantly (*p* < 0.05) smaller after the last cryoinjury and several previous cryoinjuries (Multiple), and in hypertensive (Hypertension) as well as in old (Old) rats. (**P**). In the re-endothelialized zone, the mean thickness (height) of ECs is higher than in control (no freezing) rats. (Controls for hypertensive and old rats are not shown, although these numbers are also significantly (*p* < 0.05) lower than in these experimental animals). (**Q**) Mean EC volume in the re-endothelialized zone is significantly (*p* < 0.05) lower than in control rats as well as than in the area proximally and distally from the zone of re-endothelialization (not shown). (**R**) Percentage of the EC volume occupied by mitochondria is significantly (*p* < 0.05) higher in ECs of re-endothelialized zone. (**S**) Percentage of samples containing leucocytes on the surface of ECs is significantly (*p* < 0.05) higher after multiple cryoinjuries or in sample taken from hypertensive and old rats but was not changed significantly after a single cryoinjury in normal rats. Difference between multiple-damaged, old and hypertensive animals is not significant. (**T**) Percentage of samples with leading ECs not connected with the EC front is significantly (*p* < 0.05) higher in sample obtained after multiple (Multiple) cryoinjuries and in hypertensive (DOC) rats. Scale bar in (**N**) is equal to 25 µm.

**Figure 2 biomedicines-10-02858-f002:**
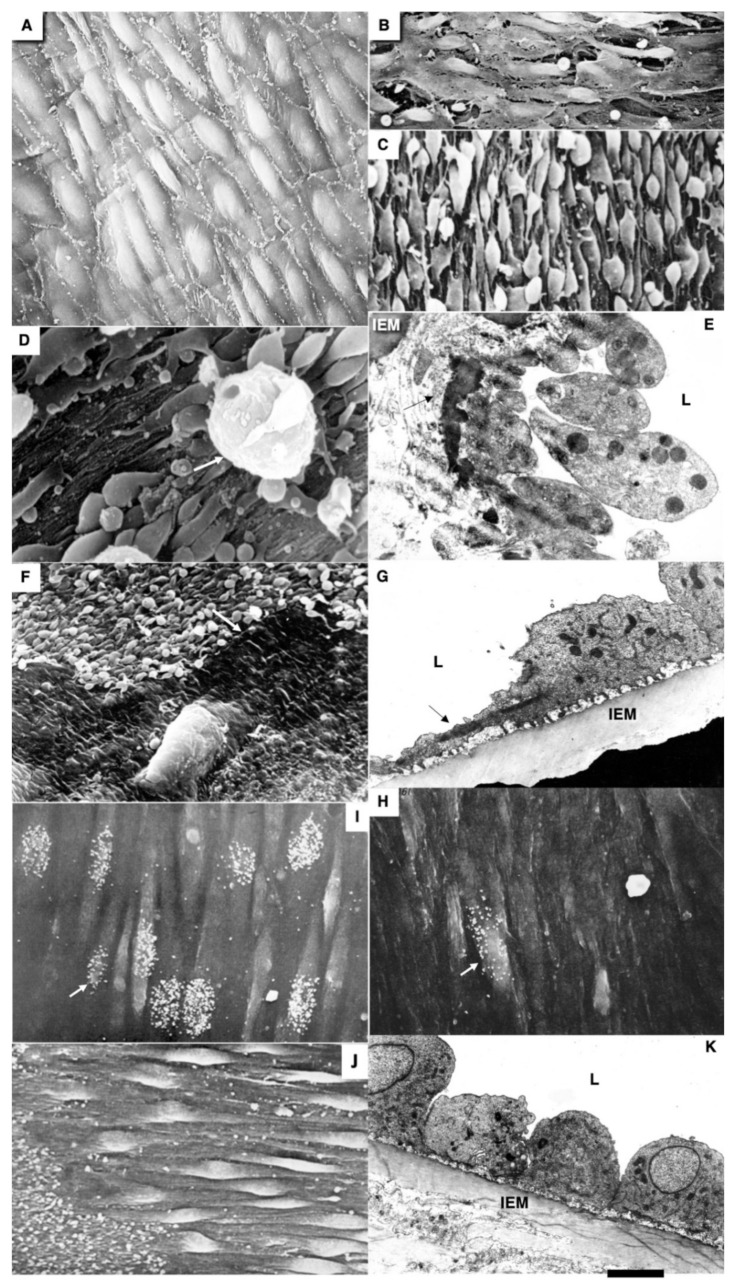
Alterations of aortic ECs after cryoinjury. (**A**) Control sample from the rat treated with the copper probe with the temperature equal to 37 °C. Normal structures of ECs. Scanning electron microscopy (SEM). (**B**) Detachment of ECs after the aortic wall freezing. SEM. (**C**) Attachment of thrombocytes to the frozen zone. SEM. (**C**) Attachment of the monolayered of platelets over the BM of ECs. SEM. (**D**) Attachment of leucocytes (arrow) and platelets. SEM. (**E**) Fine section through the platelet monolayer. Transmission electron microscopy (TEM). Arrow indicates the basement membrane and fibrin on it. L, lumen. (**F**) Lamellipodia (arrow) of the spread out EC. SEM. (**G**) Longitudinal section of the EC lamellipodia (arrow). TEM. (**H**) Control sample. Silver granules over the EC (white arrow) in S phase. SEM of radioautography. (**I**) Zone of re-endothelialization. Silver granules over the EC (white arrow) in S phase. SEM of radioautography. The re-endothelialization zone. TEM. (**J**) Leading edge of endothelial monolayer. which moves towards the left side of the image eliminating platelets (on the left). L, lumen. (**K**) Cross-section through spindle-like ECs in the zone of re-endopthelization. TEM (from Figure 1 by Mironov et al. [4]). L, lumen. Abbreviations: IEM, internal elastic membrane; L, lumen of aorta. Scale bars: 12.3 µm (**A**,**B**); 4 µm (**C**); 3 µm (**D**); 1 µm (**E**); 7 µm (**F**); 2 µm (**G**–**I**); 6 µm (**J**); 5 µm (**K**).

**Figure 3 biomedicines-10-02858-f003:**
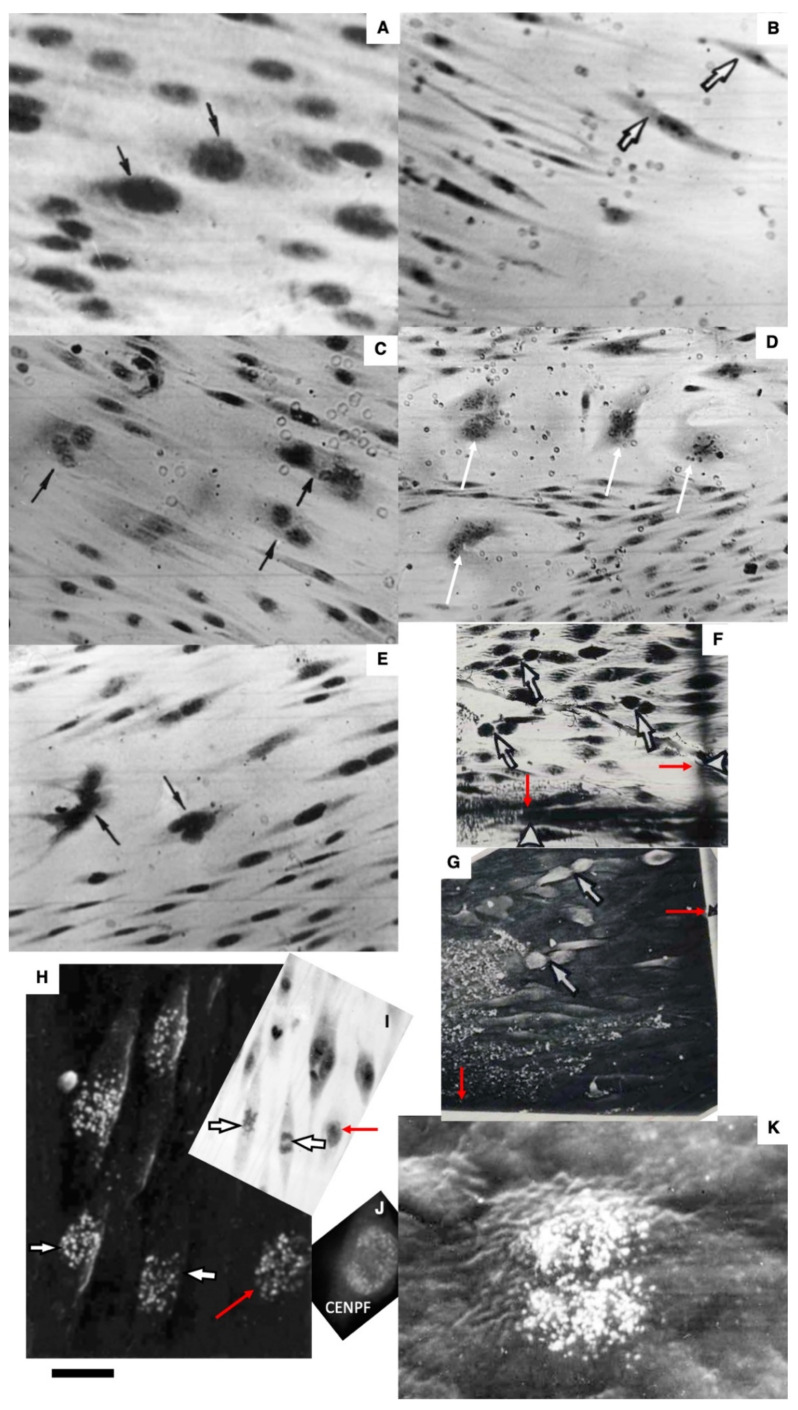
Methods of the correlative light electron microscopy (CLEM) based on en-face EC monolayers combined with SEM and the en-face analysis of aortic ECs. (**A**) Single injury. Arrows show EC in prophase. The en-face EC monolayer. (**B**) Leading ECs (arrows) moving ahead of the front of ECs. The en-face EC monolayer. Multiple injury. Leading ECs are shown by arrows (quantified in Figure 1T). (**C**) Multiple cryoinjuries of ECs. Multinuclear ECs (black arrows). The en-face EC monolayer. Quantified in Figure 5N. (**D**) The single cryoinjury in the hypertensive rats. Multinuclear ECs are shown with white arrows. (**E**) The single cryoinjury in old rat. Multinuclear ECs (black arrows). (**F**–**J**) Steps used for identification of the EC in G2 phase. (**F**,**G**) CLEM based on the combination between the en-face EC monolayer and SEM. (**F**) ECs visible in the en-face sample (**G**). The same area visible under SEM. Red arrows indicate the bars of the copper grid situated over the monolayer. White arrows show mitotic ECs. (**H**–**J**) CLEM. (**H**) Labeled EC in S, G2, and mitosis (red arrow). Red arrow shows the G2-phase EC, white arrows indicate mitotic ECs. (**I**) The same area as in (**H**) shown on the en-face EC monolayer. Red arrow shows the G2-phase EC, white arrows indicate mitotic ECs. (**J**) Labeling of the same EC for CENPF, a marker of the G2 phase. (**K**) SEM radioautography of mitotically dividing EC. Telophase. Scale bars: 14 µm (**A**); 29 µm (**B**); 25 µm (**C**,**F**,**I**); 26 µm (**E**,**G**); 12 µm (**H**); 11 µm (**J**); 3.8 µm (**K**).

**Figure 4 biomedicines-10-02858-f004:**
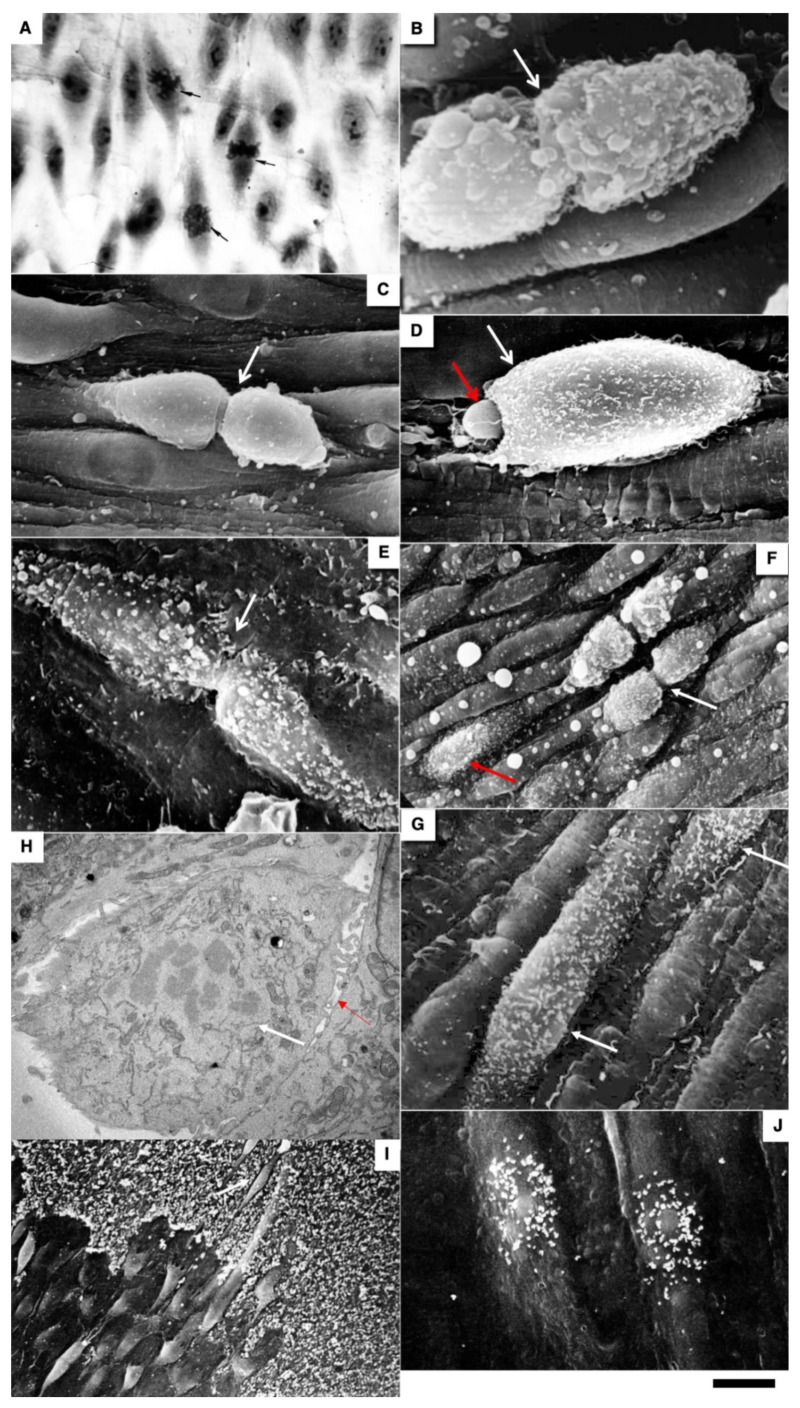
Mitotic activity of ECs during regeneration. (**A**) Flat en-face preparation of the EC monolayer. Light microscopy. Arrows show ECs in mitosis. (**B**–**G**) Different types of mitosis. Scanning electron microscopy (SEM). Mitotic ECs are shown with arrows. (**B**) EC (arrow) in early telophase. (**C**) EC (arrow) in cytokinesis. (**D**) Wide interendothelial contact (red arrow) between the mitotic EC (white arrow) and other ECs. A leucocyte inside the contact is visible (Taken from Figure 3F by Mironov et al., [4]). (**E**) EC in telophase. (**F**) Zone of re-endothelialization. Red arrow shows two ECs in cytokinesis. White arrow shows EC in metaphase. (**G**) Two ECs (arrows) in prophase. Many microvilli are visible on their surface. (**H**) Tangential section of the mitotic EC (EC in metaphase). Interendothelial contacts (red arrow) are wide. (**I)** The EC isolated from the leading endothelial edge is shows with the white arrow. (**J**) SEM radioautography demonstrates ECs in S and G2 phases. (Silver granules (white dots) indicate the localization of isotope. Scale bars: 13 µm (**A**,**I**); 3 µm (**B**); 8 µm (**C**); 3.5 µm (**D**); 4.1 µm (**E**,**G**,**J**); 10 µm (**F**); 2.8 µm (**H**).

**Figure 5 biomedicines-10-02858-f005:**
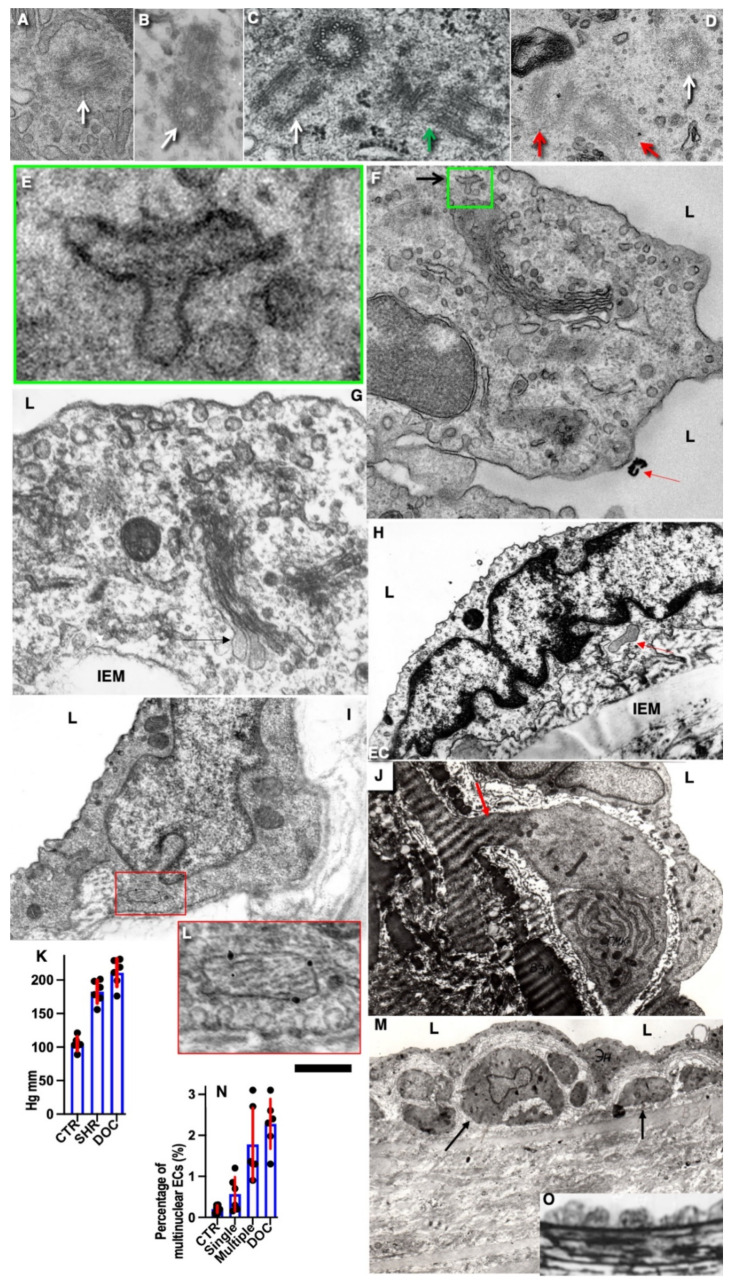
Structure of ECs during their re-endothelialization. (**A**,**B**) Centrioles (white arrows) in the G2-phase ECs. Only one pair of centrioles. (**C**) Centrioles in the S-phase EC. Two pairs of almost fully matured centrioles (white and green arrows) at a short distance from each other. (**D**) Two pairs of matured centrioles (white and red arrows) at the distance from each other equal to 500 nm or more. (**E**) Enlargement of the box indicated in green in (**F**). ERES (arrow) is visible. (**F**) Cross-section through the EC in S-phase (black silver grains is shown with red arrow) TEM section of the en-face EC monolayer. Radioautography of EC. L, lumen of aorta. (**G**) Cross EM-section of the EC. The Golgi complex with cisternal distensions (black arrow) at its trans side. L, lumen of aorta. (**H**) The cross TEM section of the EC in the G2-phase obtained after sectioning of the en-face EC monolayer. Red arrow shows the post-Golgi carrier. L, lumen of aorta. (**I**) The cross TEM section of the EC in G2 phase obtained after sectioning of the en-face EC monolayer. Membrane of post-Golgi carrier in the EC is labeled for Neu 1. L, lumen of aorta. (**J**) Migration of the medial SMC (red arrow) into intima. L, lumen of aorta. (**K**) Systolic arterial pressure in normal rats and in hypertensive rats after spontaneous hypertension (SHR) and DOC-hypertension (DOC). Arterial pressure in SHR and DOC rats is significantly (*p* < 0.05) higher than in control rats. (**L**) Enlargement tb of the red box in (**I**). (**M**) Cross-section of the regenerating SMCs (arrows). All SMCs migrate in the subendothelial space just below the thick BM. There are no SMCs in the media. L, lumen of aorta. (**N**) Percentage of randomly selected areas containing multinuclear cells is significantly (*p* < 0.05) higher in the re-endothelialized zone after multiple cryoinjuries and after a single cryoinjury in hypertensive rats. (**O**) Semithin section of the aortic wall with signs of the regeneration of SMCs. Low magnification of the area where signs of SMC migration are visible. Scale bars: 500 nm (**A**,**D**); 450 nm (**B**); 70 nm (**C**); 1200 nm (**E**); 2 µm (**F**); 650 nm (**G**); 200 nm (**I**).

**Figure 6 biomedicines-10-02858-f006:**
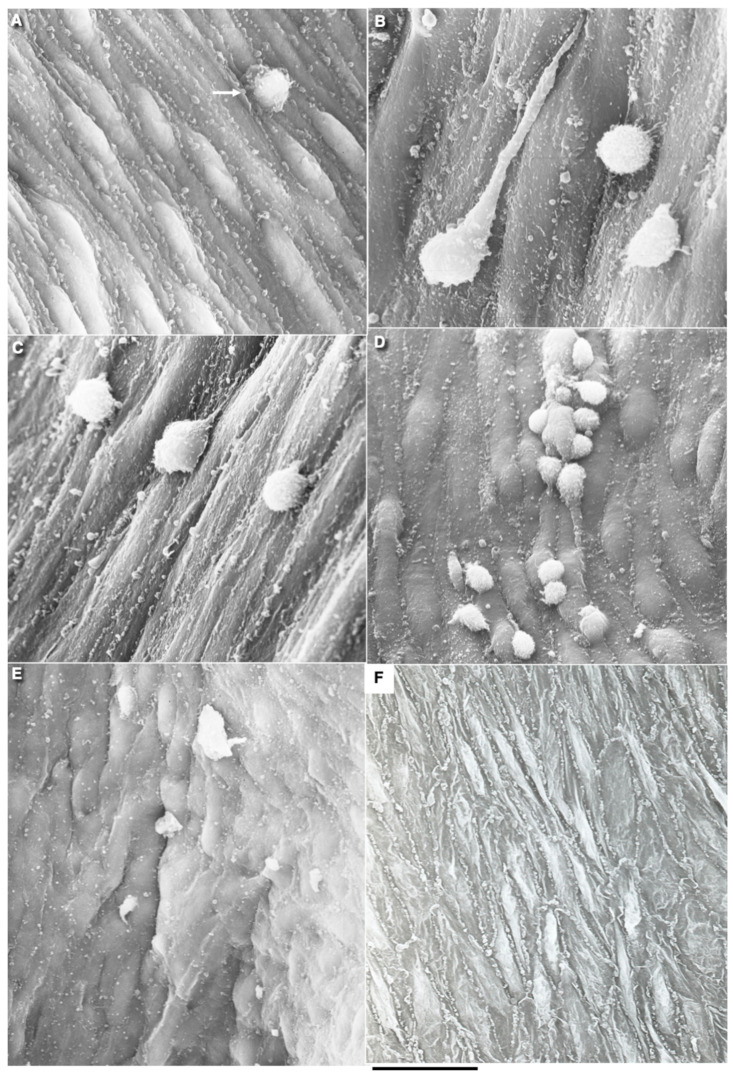
Surface of the endothelium in abdominal aorta after different influences. (**A**) Surface of ECs after the reparation of a single cryoinjury. Arrow indicates the leucocyte on the surface of ECs. (**B**) Multiple injury. (**C**) Single injury in old rats. (**D**) Single injury in hypertensive rats. (**E**,**F**) Surface of endothelium in hypertensive rats. Scale bars: 18 µm (**A**); 10 µm (**B**); 14 µm (**C**); 22 µm (**D**); 43 µm (**E**); 30 µm (**F**).

**Figure 7 biomedicines-10-02858-f007:**
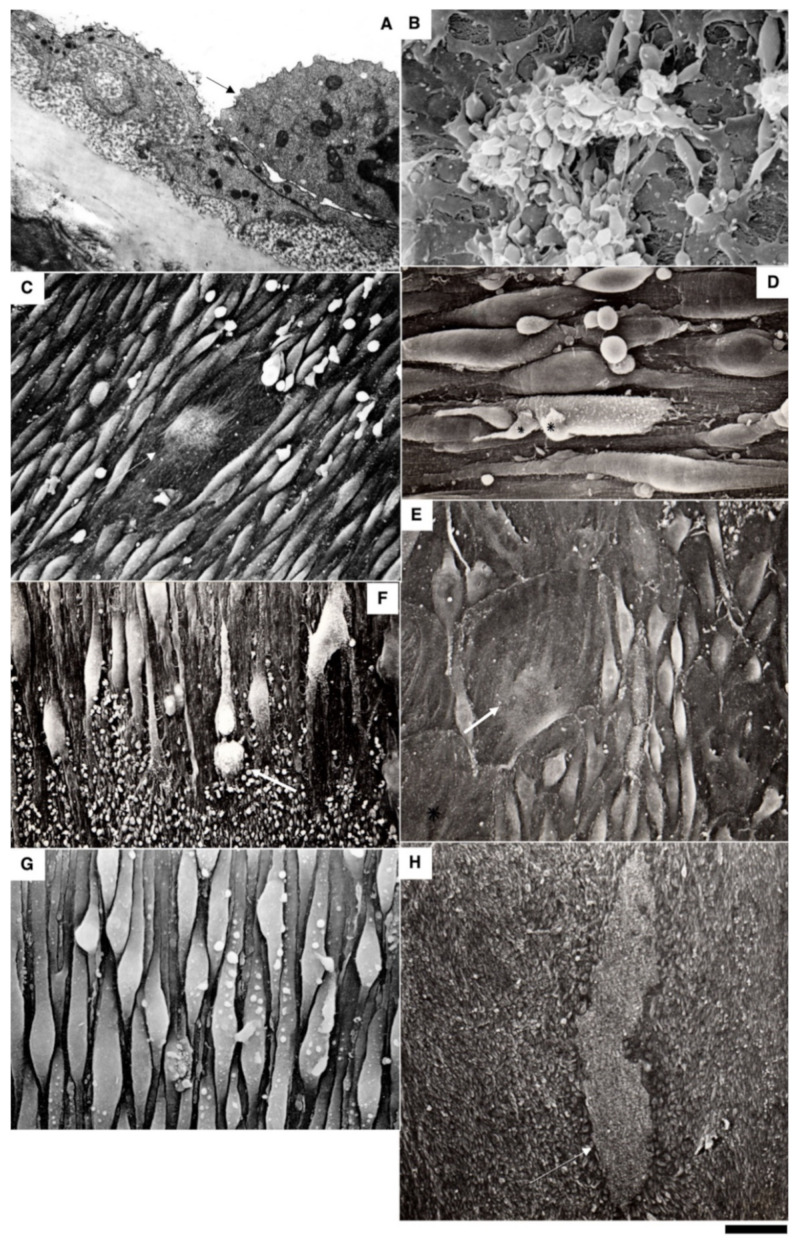
Features of ECs in re-endotheliazation zone after single (**G**) and multiple cryoinjuries (**A**,**D**,**H**), as well as after single injuries in old (**C**,**F**) and hypertensive rats (**B**,**E**). (**A**) TEM. (**B**–**H**) SEM. (**A**) Monocyte (arrow) is attached to EC of the re-endothelialized zone. (**B**) Aggregate of platelets. (**C**,**E**) Giant ECs in old © and hypertensive (**C**) rats. (**F**) Ugly ECs in the leading edge. Asymmetric mitosis (arrow). (**H**). Area not yet covered with ECs. (**H**) Altered shape of de-endothelialization zone after multiple re-endothelialization. Scale bars: 2 µm (**A**); 3 µm (**B**); 8 µm (**C**); 4 µm (**D**,**G**); 6.5 µm (**E**); 7 µm (**E**); 90 µm (**H**).

**Figure 8 biomedicines-10-02858-f008:**
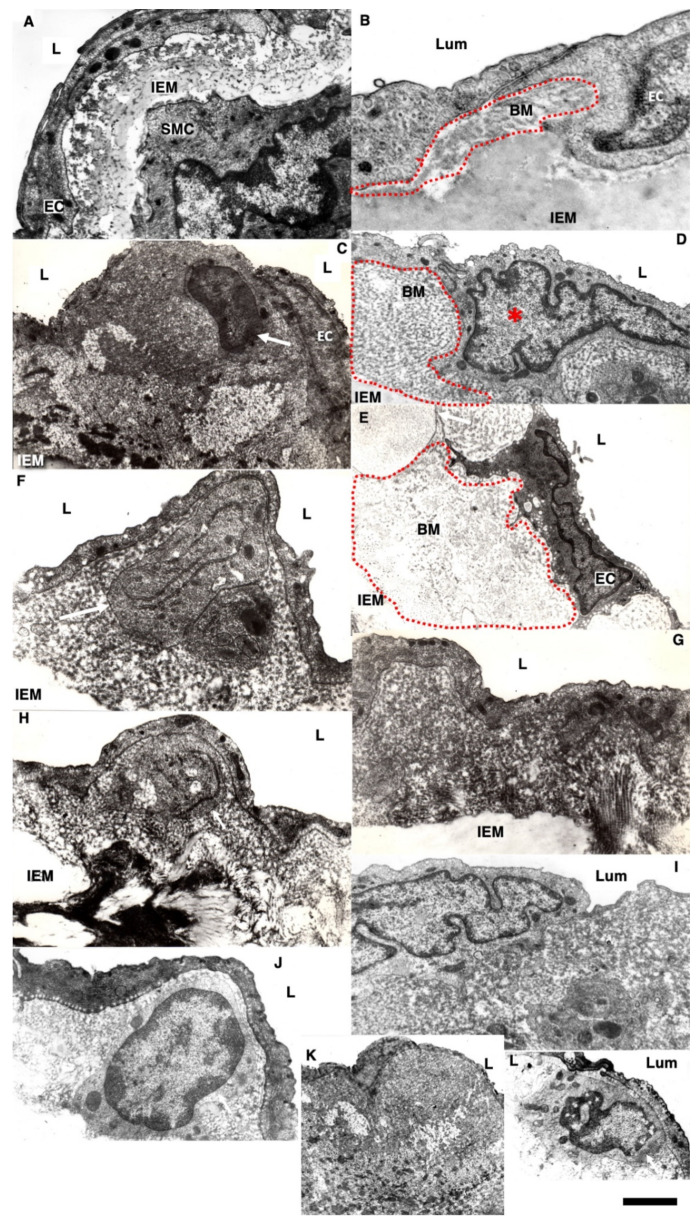
Thickness of the BM of ECs under different conditions. (**A**) Control without cryodamage (from Figure 1A by [5]; see also Figure 3A by [4]). L, lumen of aorta. (**B**) Single re-endothelialization. A few layers of BM were already visible. Lum, lumen of aorta. (**C**) BM in the hypertensive rat (from Figure 1E by [5]). L, lumen of aorta. (**D**) BM after multiple re-endothelialization (from Figure 3C by [5]). Red asterisk indicates the EC nucleus. L, lumen of aorta. (**E**,**G**,**H**) Thick subendothelial zone including BM in the 24-month-old rat (from Figure 1C by [5]). L, lumen of aorta. (**F**) Thick BM and macrophages (arrow) in intima after multiple re-endothelization (from Figure 1E by [4]). L, lumen of aorta. (**I**–**L**) Thick BM of ECs in hypertensive rats. L, lumen of aorta. (**C**,**E**,**F**,**H**,**J**,**L**) Leucocytes (white arrows) in subendothelial zone. Scale bars: 700 µm (**A**–**D**,**F**–**J**,**L**); 2.5 µm (**E**); 3 µm (**K**).

**Figure 9 biomedicines-10-02858-f009:**
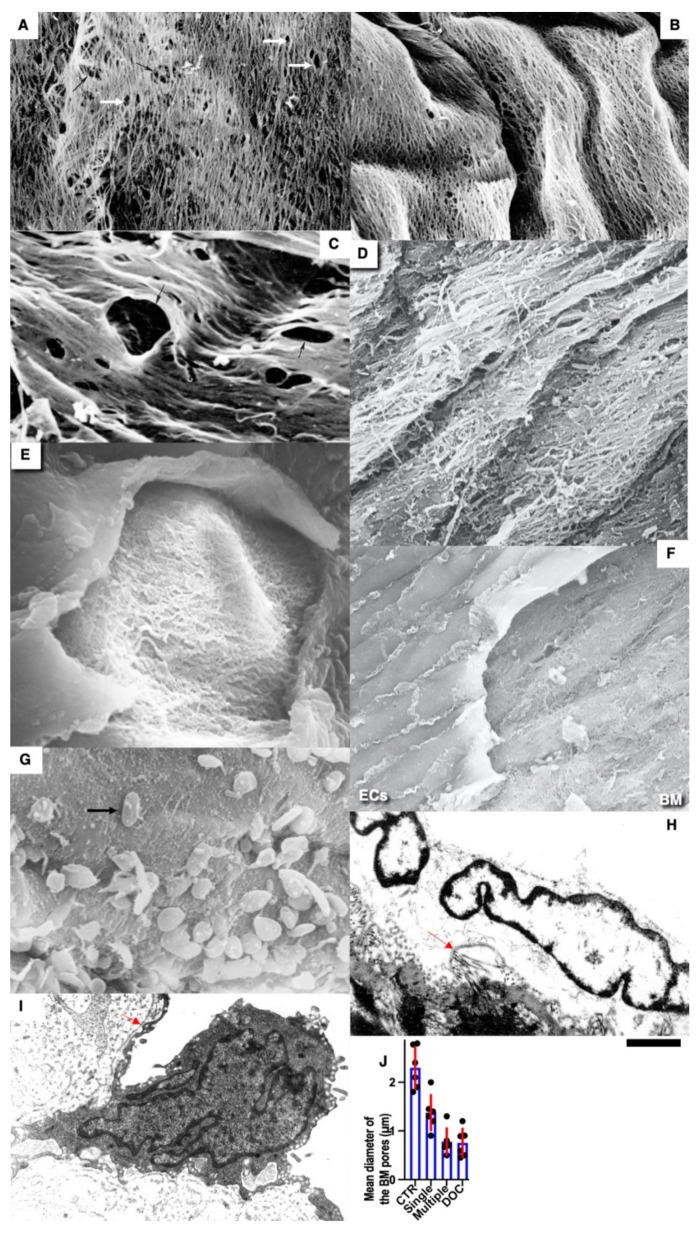
Decrease in the BM pore size after re-endothelialization. (**A**,**B**) Control. White arrows show large pores in BM, black arrows demonstrate pores in the internal elastic membrane. (**C**) Single injury. Black arrows show pores in internal elastic membrane. (**D**) Multiple injuries. (**F**) Detachment of ECs from their BM in hypertensive rats induced by freezing. SEM. (**G**) SEM of the BM pores in old rats after cryoinjuries. Arrow indicates the platelet. (**E**) Internal elastic membrane after its treatment with collagenase. (**H**,**I**) Solid BM (red arrows) is clearly visible in TEM sections (**I**) and after dissolution of lipid membranes with 1% Triton X100 (**H**). (**J**) Mean diameter of large pores in the ECBM after a single cryoinjury (“Single”), multiple cryoinjuries (“Multiple”), and after a single cryoinjury in hypertensive rats (“Doc”) is significantly (*p* < 0.05) smaller than in control rats (“CTR”). Scale bars: 7.5 µm (**A**); 8.4 µm (**B**); 4 µm (**C**); 8.1 µm (**D**); 9 µm (**E**); 15 µm (**F**); 6.5 µm (**G**); 800 nm (**H**); 1.8 µm (**I**).

**Figure 10 biomedicines-10-02858-f010:**
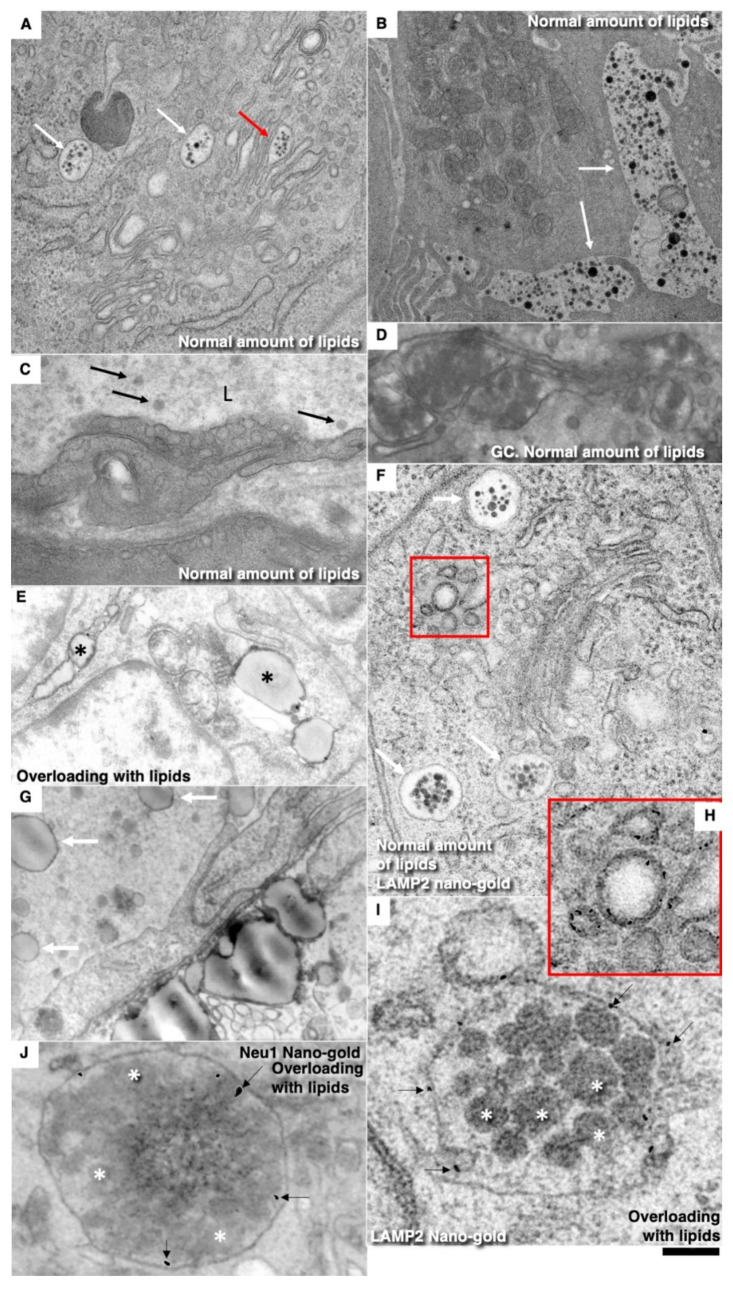
Overloading of enterocytes with lipids alters production of chylomicrons. (**A**,**B**) Control animals fed with a small amount of lipids. White arrows in (**A**) show post-Golgi carriers. Red arrows indicate the distension of the Golgi cisternae filled with normal chylomicrons. (**B**) Normal chylomicrons (white arrows) in the space between enterocytes. (**C**) Normal chylomicrons (arrows) in the lumen (L) of lymphatic capillary of the intestinal villi. (**D**) Large chylomicrons (significantly (*p* < 0.05) larger than in control rats; quantified in Figure 11I) in the distensions of the Golgi cisternae. (**E**) Large chylomicrons (black asterisks) between enterocytes in the overloaded rats. (**F**) In normally fed rats, post-Golgi carriers (white arrows) contain normal chylomicrons but did not exhibit labeling for LAMP2. (**G**) Large chylomicrons (white arrows) are visible in the lumen of the lymphatic capillary of intestinal villus. (**H**) Enlarged area indicated in the red box in (**F**). White asterisks indicate enlarged chylomicrons. (**I**,**J**) Enlarged chylomicrons (white asterisks) are present in the compartment positive for LAMP2 (**I**) and Neu1 (**J**). (Scale bars: 250 nm (**A**,**F**); 330 nm (**B**); 500 nm (**C**); 820 nm (**D**); 2500 nm (**E**); 315 nm (**G**); 105 nm (**H**); 140 nm (**I**); 300 nm (**J**).

**Figure 11 biomedicines-10-02858-f011:**
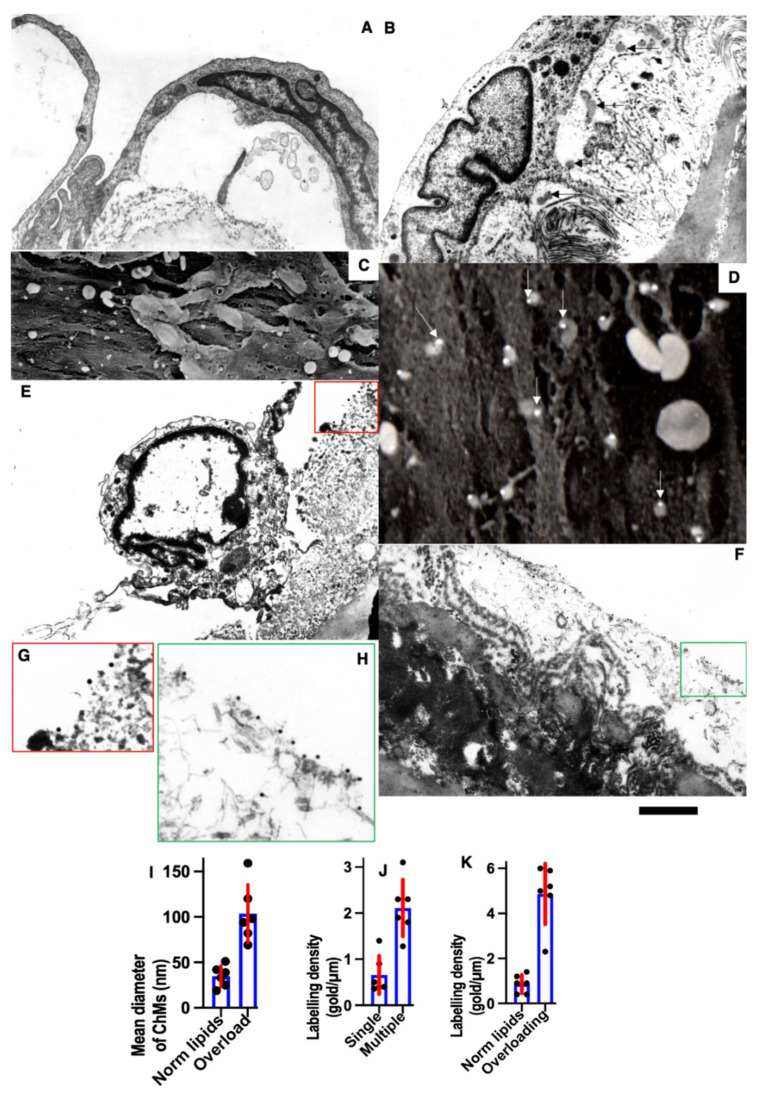
Role of extensive synthesis of the BM by ECs and overloading of enterocytes alter binding of LDLs to the BM. (**A**) Incubation with LDL obtained from the control rats fed with small amount of lipids. TEM. (**B**) Incubation of aorta with LDLs obtained from rats fed with high amount of lipids. Arrows show LDLs bound to the BM. TEM. (**C**) SEM of the nude BM incubated with LDLs obtained from normal rats. (**D**) SEM of the nude ECBM incubated with normal LDLs after multiple cryoinjuries. Higher number of LDLs labeled with gold (white arrows) are more visible than in (**C**). Quantified in (**I**). (**E**) Incubation of the nude aortic BM with the Limax flavus lectin. Attachment of gold labeled lectin to the BM is shown in (**G**) under higher magnification. (**F**) After multiple cryoinjuries, the level of the lectin attachment was higher. (**G**,**H**) Enlarged areas indicated in (**E**) and (**F**) correspondingly. Quantified in (**J**). (**I**) Overloading of enterocytes with lipids induces significant (*p* < 0.05) augmentation of the diameter of chylomicrons. (**J**) After multiple cryoinjuries, the nude BM of aortic ECs captured significantly (*p* < 0.05) more LDLs than after a single cryoinjury and consecutive re-endothelialization. (**K**) LDLs obtained from rats after their overloading with lipids (Overloading) had significantly (*p* < 0.05) higher affinity to the BM of the nude BM of the aortic BM than LDLs isolated from rats fed with lower amount of lipids (Norm lipids). Scale bars: 3.5 µm (**A**); 2.5 µm (**B**) 25 µm (**C**,**D**); 3.2 µm (**E**); 3.6 µm (**F**).

**Figure 12 biomedicines-10-02858-f012:**
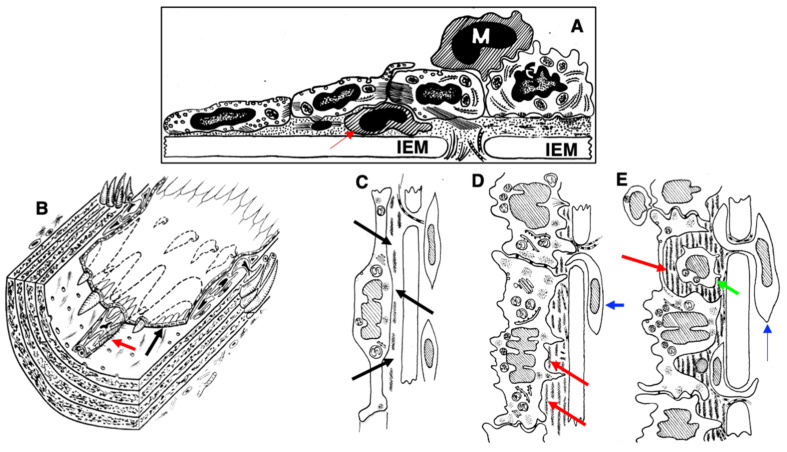
Schemes illustrating the process of EC regeneration. (**A**) Scheme demonstrating the consequence of event during endothelial regeneration after cryoinjuries. Red arrow shows leucocyte in the subendothelial space. M, monocyte. (**B**) Scheme of regeneration of SMCs in aorta after extensive cryoinjury. SMCs form spindle-like extrusion (red arrow) moving within the subendothelial layer (BM is not shown). Black arrow indicates endothelial layer. No SMCs were observed in the media. (**C**) Scheme of the normal intima in the rat aorta. BM has pores (black arrows). (**D**) Thick BM and subendothelial layer (arrow) after a single re-endothelialization. SMCs indicated with blue arrow could enter the subendothelial space only at the edges of the cryoinjury. Enlarged solid parts of BM are shown with red arrows. (**E**) Scheme of the aortic intima in hypertensive and old rats, and after multiple re-endothelialization. Pore in BM became narrow. Solid parts of BM is indicated with the red arrow. Leucocytes (green arrow) are accumulated in the subendothelial layer (from Mironov et al., [4]). SMCs indicated with blue arrow could enter the subendothelial space only at the edges of the cryoinjury.

## Data Availability

Not applicable.

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
