# Peer review of "Role of Endothelial Regeneration and Overloading of Enterocytes with Lipids in Capturing of Lipoproteins by Basement Membrane of Rat Aortic Endothelium"

_biomedicines, 2022, doi:10.3390/biomedicines10112858_

Round 1

Reviewer 1 Report

In this paper, the authors observed the migration and aggregation of leukocytes and platelets to endothelial cells and the thickening of basement membrane after cryo-injuries, as well as in hypertensive and old rats by preembedding immuno-electron microscopy assay, and investigated the occurrence time of this process.

1)     The molecular mechanism of basement membrane thickening caused by cryo-injuries are not enough.

2)     Line 58: Repeat writing “than men and men and monkeys”.

3)     In this paper, the authors focused on endothelial cell changes and basement membrane thickening after cryo-injuries, but used enterocytes during LDL lipid overloading. Why not continue to use endothelial cells.

4)     The method of determining cell cycle is not shown in this paper.

5)     Line 435: Please confirm that the age of the rat is correct“Thick BM in the 24-years old rat (from Fig. 1C by Mironov and Beznoussenko, 2022).”

6)     The description of the preembedding immuno-electron microscopy results is not clear enough. Could you describe it more specifically?

7)     There is no control group in Figure 6.

8)     There is a lot of scope for improvement in the discussion section. Authors should reconstruct key take-home points of the manuscript. The conclusion should crystallize the article and expand into future directions. If the authors read the manuscript again, the fine prints can be crystallized into a condensed, informative section.

Author Response

All comments have been taken into account, errors have been corrected and wishes have been implemented. We completely re-wrote the text. We made a special Figure dedicated to the changes of the BM. A separate drawing was made on BM and the scheme explaining our method of the correlative light electron microscopy (CLEM) based on the endothelial cell monolayer preparations. Control images are presented. Our tables have been converted into graphs and the significance of the differences is given in the Figure legends. The mistakes in the text have been corrected. The discussion has been made more focused. English was checked by and Englishman and straightened out. The reference numbers have been corrected. The methods of CLEM and the search for phases of the cell cycle are described.

Referee 1

In this paper, the authors observed the migration and aggregation of leukocytes and platelets to endothelial cells and the thickening of basement membrane after cryo-injuries, as well as in hypertensive and old rats by pre-embedding immuno-electron microscopy assay, and investigated the occurrence time of this process.

Authors: We acknowledge our referee 1 for the very useful comments. We noticed many from mistakes already after submission and. asked the editor to cancel our first submission. However, the text was already sent to referees. The control experiments were presented in our previous papers and therefore we decided to avoid them. Now this was corrected. We included additional experiments with the en-face EC monolayers, scanning microscopy, and correlated light electron microscopy.

1)     The molecular mechanism of basement membrane thickening caused by cryo-injuries are not enough.

Authors: We added additional information about thickening. Molecular mechanisms are the same which stimulate ECs to migrate and then divide. In the text, we indicated these mechanisms. In our paper submitted we demonstrated that secretion of BM components also in culture occurs during G2 phase.

2)     Line 58: Repeat writing “than men and men and monkeys”.

Authors: We corrected this mistake.

3)     In this paper, the authors focused on endothelial cell changes and basement membrane thickening after cryo-injuries, but used enterocytes during LDL lipid overloading. Why not continue to use endothelial cells.

Authors: It is not completely clear what the referee means. If to use the culture of ECs in culture the answer would be the following. In culture there is no blood with its cells, pulsation of blood pressure, perforated BM. In culture BM does not have pores which are present in aorta as organ. The natures of BM formed in culture would be different. In future we plan to confirm our data using cells culture, but these data would be too large for the inclusion in the current paper.

4)     The method of determining cell cycle is not shown in this paper.

Authors: We rewrote this part of the manuscript and made it clearer.

5)     Line 435: Please confirm that the age of the rat is correct “Thick BM in the 24-years old rat (from Fig. 1C by Mironov and Beznoussenko, 2022).”

Authors: We corrected this mistake. Actually, there should 24 months.

6)     The description of the pre-embedding immuno-electron microscopy results is not clear enough. Could you describe it more specifically?

Authors: We rewrote this part and hope that now the description is much clearer

7)     There is no control group in Figure 6.

Authors: We added the control group.

8)     There is a lot of scope for improvement in the discussion section. Authors should reconstruct key take-home points of the manuscript. The conclusion should crystallize the article and expand into future directions. If the authors read the manuscript again, the fine prints can be crystallized into a condensed, informative section.

Authors: We rewrote the discussion part.

Reviewer 2 Report

In this study, the authors apply freezing damage to the endothelial cell layer of aortas of rat to look for their regeneration behavior in different contexts (single or repeated damage, hypertensive or aged background). In addition, the authors study the ability of previously damaged and regenerated areas of the aorta to capture lipoproteins to the basement membrane in the context of overloading of enterocytes with lipids. Thereby, the data support the Response-to-Injury hypothesis for development of atherosclerosis.

The study is mainly focusing on effects observed in electron microscopy images. Unfortunately, frequently the images of treated animals are shown without corresponding control images making it difficult to judge on the effects described in the text and also to clearly recognize these effects within the images. Some image could be better labeled and figure descriptions could be expanded (especially Fig 1, 2 and 4) so that unexperienced readers are better guided through the images. Some of the images of Figure 2 and 5 were reused from previous publications of the authors which is declared within the figure legend.

Instead of graphs, the authors frequently use tables to presents their quantified data. Since the significant differences are not indicated in these tables, it is not possible to easily judge what parameters were statistically compared with each other. In the statistics part of the methods the authors declare words of the text representing indications for statistical differences, however, it is too much effort to evaluate each of the observed effects that way. In addition, there are often multiple groups compared in the tables and is not clear what statistical tests were applied to each data set disallowing any judgement on that data.

The manuscript has many citations and a very detailed and informative discussions. However the discussion is not focused on the author’s data and would better fit in a review article written that way. In addition, some of the author’s data discussed there lack respective controls and therefore statements in the discussion must be weakened.

Major corrections:

Line 23 The term oil is less well defined, please use the word lipids instead

Lines 148, 361: oil solution is not a very well defined term. Please precise its origin or its composition.

Figure 1A: It is hard to judge if EC detachment took place due to freezing of the aorta or preparation of the aorta for electron microscopy if no pictures of untreated controls are shown in a similar fashion.

Figure 1D, G, F: Can the authors indicate the location of deendothelialized area and intact endothelial area within their images.

Figure 1I: From the text in the results section, it is completely unclear what is displayed in this diagram. Please describe it at the respective position in the text.  First after reading further, the reader is related to Fig 1I as representing the area of endothelial regeneration in the respective conditions. The Y-axes of the graph is not centered. Statistical significances are unclear and tests are not mentioned.

Figure 3: Untreated control images of centrioles of non-reendothelialized regions or control animals would be nice

Figure 4D: The visible asterisks are not explained in the figure legend

Figure 5. Since it is a comparison of the thickness of the basement membrane, please indicate the location of the basement membrane in each image.

Figure 6A. No control images of normal sized chylomicrons under control conditions are shown and the size is also not quantified so its unclear if chylomicrons are really enlarged

Figure 6N-P: No LDL data of untreated controls are shown. Statistical significances are unclear and tests are not mentioned.

Line 463 : “Reendothelization induced thickening of the network-liker BM of aortic ECs and dis-463 appearance of its pores.” Where is the quantification of reduction in pores?

Line 479: “In our hands, after reendothelization, the affinity of BM to LDLs became higher than the normal rats.” It is not possible to state that because no data of normal rat control (untreated) is shown in the Figure 6N.

Data in all tables: please indicate location of significant differences and applied tests if data were statistically analysed

Minor corrections

correct  “endothelization” to “endothelialization” throughout the manuscript. Please uniformly use either the form “reendothelialization” or “re-endothelialization” throughout the manuscript.

Line 29-34 This sentence is interrupted by two sub-sentences within brackets which makes it very difficult to get the information. Please split in at least two sentences.

Line 70 In terms of blood pressure, what is meant with “single load”? Please better define.

Line 19 add a comma after the word re-endothelialization

Lines 22, 139, 238, 268, 271, 306, 307, 328, 338, 374, 388, 402, 419, 428, 440, 441, 454, 458, 465, 479, 483, 496, 535, 565: correct the a double space between two words

Line 60 … than men and men and monkeys … is there an accidently doubling of men?

Line 108-109 please check this sentence for its meaning

Line 143 the word “either” indicates a following “or” which does not occur in the sentence

Line 150 change “DOCK”  to  “DOC”

Line 171 change “was” to “were”

Line 172 copper instead of coper

Line 228 Define abbreviation ERES

Line 239 Define abbreviation SEM

Line 252 Insert space between pH and 5.8

Line 263 use subscript for numbers in these chemicals

Line 277 Define abbreviation CLEM

Line 307 add the word “to” to the BM of ECs

Line 311 ECs had a characteristic shape with sharp … in the sentences, there is missing what is sharp (with sharp what?).

Line 312 On cross sections, ECs exhibited ovoid shape à please indicate that this is Fig. 1H

Line 366 Define abbreviation LAMP2, was this an antibody staining? If that is the case, the antibody is not listed in material & methods

Line 367 Define abbreviation BLPM

Line 377 dot should be comma

Line 390: Sentence is strange

Line 398 Define abbreviation TEM

Table 1: Please define DOKA in the methods as it is unclear from the tables

Line 406 should à shown

Line 463: liker à like?

488 remove dot

Author Response

All comments have been taken into account, errors have been corrected and wishes have been implemented. We completely re-wrote the text. We made a special Figure dedicated to the changes of the BM. A separate drawing was made on BM and the scheme explaining our method of the correlative light electron microscopy (CLEM) based on the endothelial cell monolayer preparations. Control images are presented. Our tables have been converted into graphs and the significance of the differences is given in the Figure legends. The mistakes in the text have been corrected. The discussion has been made more focused. English was checked by and Englishman and straightened out. The reference numbers have been corrected. The methods of CLEM and the search for phases of the cell cycle are described.

In this study, the authors apply freezing damage to the endothelial cell layer of aortas of rat to look for their regeneration behavior in different contexts (single or repeated damage, hypertensive or aged background). In addition, the authors study the ability of previously damaged and regenerated areas of the aorta to capture lipoproteins to the basement membrane in the context of overloading of enterocytes with lipids. Thereby, the data support the Response-to-Injury hypothesis for development of atherosclerosis.

Authors: We acknowledge our referee 2 for the very useful comments. We noticed many from mistakes already after submission and. asked the editor to cancel our first submission. However, the text was already sent to referees.

The study is mainly focusing on effects observed in electron microscopy images. Unfortunately, frequently the images of treated animals are shown without corresponding control images making it difficult to judge on the effects described in the text and also to clearly recognize these effects within the images.

Authors: We added control images. There is no EC peeling in the control samples. There is no peeling in the zones adjacent to the freezing area, which proves that the AK are peeling off after the formation of ice crystals. This has been proven in 1977 by Malczak and Buck (1977), and then by our numerous works (see papers in the reference list by the author: Rekhter). Our images published previously demonstrated that ECs are peeling off due to the formation of ice crystals and perforation of the EC PM.

Some image could be better labeled and figure descriptions could be expanded (especially Fig 1, 2 and 4) so that unexperienced readers are better guided through the images. Some of the images of Figure 2 and 5 were reused from previous publications of the authors which is declared within the figure legend.

Authors: We added additional arrows and indications.

Instead of graphs, the authors frequently use tables to presents their quantified data. Since the significant differences are not indicated in these tables, it is not possible to easily judge what parameters were statistically compared with each other.

Authors: We exclude Table and included graphs with the indications of statistical differences.

In the statistics part of the methods the authors declare words of the text representing indications for statistical differences, however, it is too much effort to evaluate each of the observed effects that way. In addition, there are often multiple groups compared in the tables and is not clear what statistical tests were applied to each data set disallowing any judgement on that data.

Authors: We explained better our statistics; included separated groups and used graphs instead of Tables.

The manuscript has many citations and a very detailed and informative discussions. However the discussion is not focused on the author’s data and would better fit in a review article written that way.

Authors: We included also data by other authors and made discussion shorter and more focused on our data.

In addition, some of the author’s data discussed there, lack respective controls and therefore statements in the discussion must be weakened.

Authors: We added control images and statistics into the text.

 Major corrections:

Line 23 The term oil is less well defined, please use the word lipids instead

Authors: We replaced this term in the text.

Lines 148, 361: oil solution is not a very well defined term. Please precise its origin or its composition.

Authors: We added the description and quate our previous paper.

Figure 1A: It is hard to judge if EC detachment took place due to freezing of the aorta or preparation of the aorta for electron microscopy if no pictures of untreated controls are shown in a similar fashion.

Authors: We added control images.

Figure 1D, G, F: Can the authors indicate the location of deendothelialized area and intact endothelial area within their images.

Authors: We showed these areas.

Figure 1I: From the text in the results section, it is completely unclear what is displayed in this diagram. Please describe it at the respective position in the text.  First after reading further, the reader is related to Fig 1I as representing the area of endothelial regeneration in the respective conditions. The Y-axes of the graph is not centered. Statistical significances are unclear and tests are not mentioned.

Authors: We changed the text and the image according to the recommendation. We included significance in the Figure legend.

Figure 3: Untreated control images of centrioles of non-reendothelialized regions or control animals would be nice.

Authors: We added control images.

Figure 4D: The visible asterisks are not explained in the figure legend

Authors: We explained this.

Figure 5. Since it is a comparison of the thickness of the basement membrane, please indicate the location of the basement membrane (BM) in each image.

Authors: We indicated the location of the BM

Figure 6A. No control images of normal sized chylomicrons under control conditions are shown and the size is also not quantified so it is unclear if chylomicrons are really enlarged

Authors: We added these data.

Figure 6N-P: No LDL data of untreated controls are shown. Statistical significances are unclear and tests are not mentioned.

Authors: We added these data.

Line 463 : “Reendothelization induced thickening of the network-liker BM of aortic ECs and dis-463 appearance of its pores.” Where is the quantification of reduction in pores?

Authors: We added this measurement and presented the graph. Also we added the graph with the mean diameters of chylomicrons visible between enterocytes after normal feeding of lipid and overloading.

Line 479: “In our hands, after reendothelization, the affinity of BM to LDLs became higher than the normal rats.” It is not possible to state that because no data of normal rat control (untreated) is shown in the Figure 6N.

Authors: We corrected this.

Data in all tables: please indicate location of significant differences and applied tests if data were statistically analyzed.

Authors: We made this.

 Minor corrections

Correct  “endothelization” to “endothelialization” throughout the manuscript. Please uniformly use either the form “reendothelialization” or “re-endothelialization” throughout the manuscript.

Authors: We replaced this term.

Line 29-34 This sentence is interrupted by two sub-sentences within brackets which makes it very difficult to get the information. Please split in at least two sentences.

Authors: We corrected this sentence.

Line 70 In terms of blood pressure, what is meant with “single load”? Please better define.

Authors: We explained this better.

Line 19 add a comma after the word re-endothelialization.

Authors: We added this.

Lines 22, 139, 238, 268, 271, 306, 307, 328, 338, 374, 388, 402, 419, 428, 440, 441, 454, 458, 465, 479, 483, 496, 535, 565: correct the double space between two words.

Authors: We corrected this.

Line 60 … than men and men and monkeys … is there an accidently doubling of men?

Authors: We corrected this.

Line 108-109 please check this sentence for its meaning

Authors: We corrected the sentence.

Line 143 the word “either” indicates a following “or” which does not occur in the sentence.

Authors: We changed the sentence.

Line 150 change “DOCK”  to  “DOC”

Authors: We corrected this word.

Line 171 change “was” to “were”

Authors: We corrected the sentence.

Line 172 copper instead of coper

Authors: We corrected this word.

Line 228 Define abbreviation ERES

Authors: We eliminated this abbreviation.

Line 239 Define abbreviation SEM

Authors: We defined this abbreviation

Line 252 Insert space between pH and 5.8

Authors: We inserted space.

Line 263 use subscript for numbers in these chemicals

Authors: We corrected this.

Line 277 Define abbreviation CLEM

Authors: We eliminated this abbreviation.

Line 307 add the word “to” to the BM of ECs

Authors: We added this word.

Line 311 ECs had a characteristic shape with sharp … in the sentences, there is missing what is sharp (with sharp what?).

Authors: We corrected the sentence.

Line 312 On cross sections, ECs exhibited ovoid shape  please indicate that this is Fig. 1H.

Authors: We indicated this.

Line 366 Define abbreviation LAMP2, was this an antibody staining? If that is the case, the antibody is not listed in material & methods

Authors: We defined this abbreviation.

Line 367 Define abbreviation BLPM

Authors: We eliminated this abbreviation.

Line 377 dot should be comma

Authors: We corrected this.

Line 390: Sentence is strange

Authors: We corrected this sentence.

Line 398 Define abbreviation TEM

Authors: We defined this abbreviation.

Table 1: Please define DOKA in the methods as it is unclear from the tables.

Authors: We defined this.

Line 406 should  shown

Authors: We corrected this.

Line 463: liker  like?

Authors: We corrected this.

488 remove dot

Authors: We corrected this.

Round 2

Reviewer 2 Report

Revised manuscript is now OK.